# Soil Microbial Community Composition and Diversity Analysis under Different Land Use Patterns in Taojia River Basin

**Zhe He** [1], **Chenglin Yuan** [1], **Peirou Chen** [1], **Ziqiang Rong** [1], **Ting Peng** [1], **Taimoor Hassan Farooq** [2], **Guangjun Wang** [1], **Wende Yan** [1] **and Jun Wang** [1,*]

1 National Engineering Laboratory for Applied Technology in Forestry and Ecology in South China, Central South University of Forestry and Technology, Changsha 410004, China; hezhe011006@gmail.com (Z.H.); yuanchenglin0424@gmail.com (C.Y.); chenpr2021@gmail.com (P.C.); rongziqiang666666@gmail.com (Z.R.); zlhuang1103@gmail.com (T.P.); guangjunwang@csuft.edu.cn (G.W.); t20001421@csuft.edu.cn (W.Y.)

2 Bangor College China, A Joint School between Bangor University and Central South University of Forestry and Technology, Changsha 410004, China; t.farooq@bangor.ac.uk

* Correspondence: jwang0829@csuft.edu.cn; Tel./Fax: +86-731-85623868

**Abstract:** Soil microorganisms are greatly affected by their microenvironment. To reveal the influence of different land use patterns on the composition and diversity of soil bacterial and fungal communities, this study analyzed microbial (bacteria and fungi) community composition and diversity under different land use patterns (vegetable land, wasteland, woodland, cultivated land) based on 16S rRNA, 18S rRNA, and high-throughput sequencing method in the Taojia River Basin. Spearman analysis and redundancy analysis (RDA) were used to explore the correlation between soil physicochemical properties and soil fungal and bacterial community composition, and a partial least squares path model (PLS-PM) was constructed to express the causal relationship between soil physicochemical properties and soil bacterial and fungal community diversity. The results showed that the soil bacterial species richness was highest in vegetable land and the lowest in the wasteland. *Proteobacteria* is the dominant phylum (20.69%–32.70%), and *Actinobacteria* is the dominant class (7.99%–16.95%). The species richness of fungi in woodland was the highest, while was the lowest in cultivated land. The dominant phylum of fungi in vegetable land, woodland, and cultivated land is *Mucoromycota*, 29.39%, 41.36%, and 22.67%, respectively. *Ascomycota* (42.16%) is the dominant phylum in wasteland. *Sordariomyetes* of *Ascomycota* is the dominant class in wasteland and cultivated land. *Mortierellomycetes* and *Glomeromycetes* of *Mucoromycota* are the dominant class in vegetable land and woodland. The results of the Spearman analysis revealed that the dominant groups in the bacterial and fungal communities had significant correlations with soil pH, clay, and sand ($p < 0.01$). The RDA results showed that soil clay, pH, and moisture were the key environmental factors affecting the diversity of soil microbial communities. Fungal diversity is more affected by different land use patterns than bacteria. These results provided a theoretical basis for the changes in soil microbial community composition and diversity in river basins.

**Keywords:** Taojia River Basin; land use; microbial community composition; diversity index





## 1. Introduction

Soil microorganisms are an important part of the soil ecosystem and play an important role in decomposing and forming soil minerals and organic matter, promoting nutrient cycling and transformation, aggregates formation, and soil fertility succession [1–3]. Soil microorganisms are also the most active biological factors in the formation of soil aggregates [4]. Studies by Na et al. showed that fungi wrapped microaggregates into large aggregates through extraradical hyphae, eliminated the spatial limitation of microaggregate formation, and secreted a large amount of hyphal secretions directly or indirectly affecting status of soil aggregates [5–7]. Since soil microorganisms are extremely sensitive

to changes in soil environment, changes in the soil microbial community structure can reflect changes in soil biological activity and the quality of soil environment [8,9]. Thus, the characteristics of the soil microbial community structure can be used as an important index to detect soil quality and stability [10]. Under the same natural factors, soil microbial community composition and diversity are largely influenced by land use patterns.

Changes in land use patterns directly affect soil physicochemical properties, thereby affecting the diversity of soil microbial community structure. Reclamation and tillage management practices changed the vegetation cover and soil physicochemical properties, which were closely related to soil environmental heterogeneity and are important factors driving soil microbial communities [11]. Wang Xiaohan et al. confirmed that the orderly growth and diversity of soil microbial communities are related to the development of soil fertility, and the succession of soil microbial communities and diversity to high-quality soil after tidal flat reclamation [12]. Ji Hongyi et al. found that soil microbial composition was greatly influenced by soil type (Adonis, $p < 0.001$) and soil use patterns (Adonis, $p < 0.01$), and the interaction patterns of soil microbial communities in different soil types were different [13]. Li Yanlin et al. found that there were significant differences in bacterial diversity and community distribution among different land use patterns in the Yangtze River Basin of Chongqing. Moreover, the sequence of changes in bacterial diversity was consistent with changes in soil physical and chemical properties. Microbial diversity analysis can be achieved by using alpha diversity, which responds to the diversity and richness of species within a single sample [14]. The commonly used indices are the Shannon index, Simpson's index, observed species index, and Chao1 index. The number, diversity, and community distribution of bacteria were significantly correlated with soil moisture content, pH, and soil enzyme activity [15]. Li Jinbiao et al. found that the change of land use type had a significant impact on the composition and diversity of soil bacteria community by changing soil chemistry in desert oasis ecotone. The chemical properties affected by land use type are mainly soil organic carbon content, followed by total nitrogen, total phosphorus, total potassium, and pH [16]. Yining et al. found, in a study of microbial community composition and function in swamp wetland, meadow wetland, forest land, farmland, and alkaline land, that land use type determines all aspects of soil microbial community, including the highest bacterial diversity and fungal community richness index in forest land, and soil parameters and total nitrogen significantly affect the abundance and diversity of the soil microbial community [17]. Many studies have been carried out at home and abroad on the effects of anthropogenic activities on soil microbial community composition and diversity, such as agriculture, urban development, and pesticide use [18,19]. However, there are few reports on the effects of different land use patterns on soil microbial diversity and community structure in river basins.

The Taojia River is a secondary tributary of the Xiangjiang River, located in the southwest of Chenzhou City, Hunan Province, with a total length of about 58.5 km and a total area of 602 square kilometers in its basin [20]. The long-term mining and beneficiation process in the upper reaches of the Taojia River has caused serious soil erosion, sediment accumulation, and heavy metal pollution, and the pollution of production and life on the riverbank has made the non-point source pollution of the river basin increasingly severe. By 2021, the average soil concentrations of As, Pb, and Zn pollution in the Taojia River Basin are 457, 373, and 387 mg/kg, respectively, while the concentration of Cd is 1.91 mg/kg [21]. In view of this, this study used high-throughput sequencing methods to analyze and compare the bacterial and fungal community composition and diversity as well as their related environmental factors under different land use patterns in the Taojia River Basin. The main purposes of this study were: (1) to investigate the response of soil microbial diversity to different land use patterns; (2) to reveal the microbial community composition and soil physicochemical properties under different land use patterns; (3) to evaluate the influence of soil physicochemical properties on microbial community composition and diversity. This study will provide a theoretical basis for the changes in soil microbial community composition and diversity in river basins and has important practical implications for

the comprehensive analysis of soil microbial diversity and the implementation of land use management tailored to the characteristics of the Taojia River Basin, including soil ecological restoration and maintenance of sustainable land use.

## 2. Materials and Methods

### 2.1. Location of the Study Area

The study area is located in Xinmadi Village, Fangyuan Town, Guiyang County, Hunan Province (112°13′–112°55′ E, 25°27′–13°6′ N), which belongs to a subtropical humid monsoon climate, with four distinct seasons, abundant precipitation, an annual average temperature of about 18 °C, an annual average rainfall of about 1490 mm, and a rainy season of July to September [20]. The soil is mainly Ultisol, Oxisol, and Alfisol [22] In winter and early spring, farmers work and sow crops in their vegetable land. The woodland area of the Taojia River Basin accounts for more than 50% of the total area, mainly wild forest. The cultivated land mainly grows crops including wheat, corn, soybean and so on. Wasteland is often used for livestock farming or infrastructure construction (Figure 1).

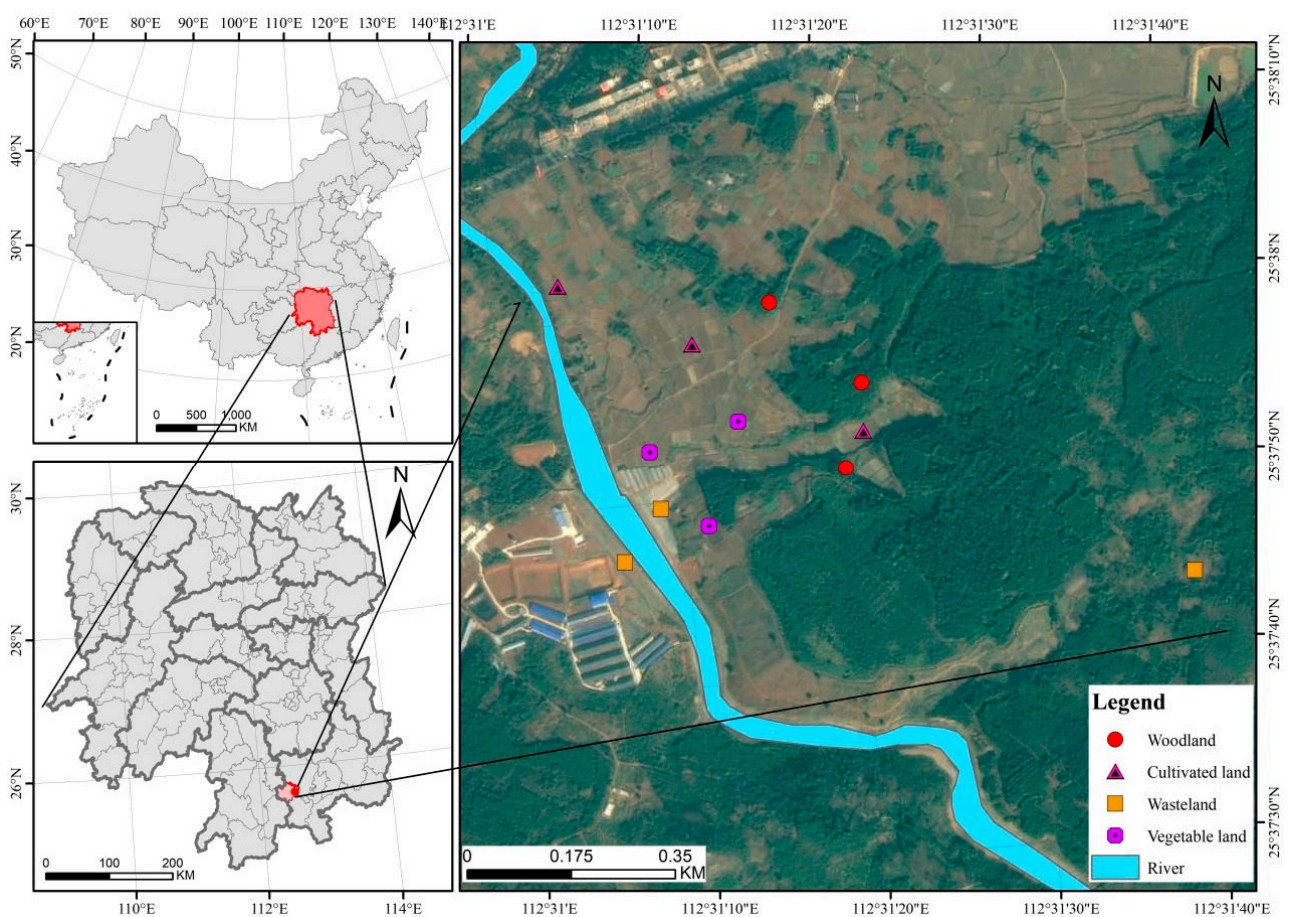

**Figure 1.** Study area and sampling site.

### 2.2. Sampling

Sampling will be conducted in the spring in March 2021. In the study area with a similar slope and consistent aspect, four land use patterns of vegetable land (VL), wasteland (WL1), woodland (WL2), and cultivated land (CL) in the Taojia River Basin, a tributary of Xiangjiang River, Guiyang County, Chenzhou City, Hunan Province, were selected as experimental plots, and three replicates of each pattern were sampled. Three 20 × 20 m plots were set up for each land use type, and 0–10 cm soil samples were collected according to an "S" shape using soil drills in each plot, mixed and placed in a sterile Ziploc bag. In

total, 12 soil mixed samples were collected to study the effects of different land use patterns on soil microbial community composition and diversity.

### 2.3. Determination of Soil Physicochemical Properties

Soil textural composition, soil pH, soil organic matter content, and soil total nitrogen content were analyzed by laser particle size analyzer, glass electrode method, potassium dichromate volumetric method, and Kjeldahl nitrogen method [23–26]. The instruments involved in this paper include PHS-3E, automatic Kjeldahl tester, and microwave digestion instrument.

### 2.4. Total Soil DNA Extraction, PCR Amplification, and Sequencing

The total soil DNA was extracted by OMEGA E.Z.N.A.TM Mag-Bind Soil DNA Kit, and the extracted genomic DNA was detected by 1% agarose gel electrophoresis. The V4 variable region of bacterial 16S rRNA gene was amplified by PCR. The bacterial primer sequences were 515F and 806R [27,28]. (515F: 5′-GTGCCAGCMGCCGCGG-3′; 806R: 5′-GGACTACHVGGGTWTCTAAT-3′). PCR amplification system (50 μL) [29]: 2 × Taq master Mix 25 μL, Bar-PCR upstream and downstream primers (10 μmol/L) 1 μL, template DNA 1 μL, ddH2O to 50 μL. Each sample was amplified in triplicate with the 50 μL reaction under the following conditions [30]: 30 cycles of denaturation at 94 °C for 30 s, annealing at 55 °C for 30 s and extension at 72 °C for 30 s; with a final extension at 72 °C for 10 min. The fungal 18S rRNA gene variable region fragment was amplified by PCR, and the fungal primer sequence were NS1 and GCFung [30,31]. NS1 (5′-GTAGTCATATGCTTGTCTC-3′) GCFung (5′-GC clamp-ATTCCCCGTTACCCGTTG-3′). PCR amplification system (50 μL) [32]: 2 × Taq master Mix 25 μL, Bar-PCR upstream and downstream primers (10 μmol/L): 1 μL; template DNA: 1 μL; ddH2O: 50 μL. Each sample was amplified in triplicate with the 50 μL reaction mixtures under the following conditions [33]: 30 cycles of denaturation at 94 °C for 30 s, annealing at 58 °C for 30 s and extension at 72 °C for 30 s, with a final extension at 72 °C for 10 min.

Under the same conditions, the samples were amplified by the Pfu high-fidelity DNA polymerase of TransGen Biotech, and the PCR products were recovered by 0.8-fold magnetic beads (Vazyme VAHTSTM DNA Clean Beads) through shaking, adsorption, and elution. The PCR amplification-recovered products were quantified using a microplate reader (BioTek, FLx800) quantification instrument of Berten in combination with Quant-iT PicoGreen dsDNA Assay Kit fluorescent agent. The Illumina TruSeq Nano DNA LT Library Prep Kit was used to prepare the sequencing library. The highlight base at the 5′ end of the amplified DNA sequence was excised, a phosphate group was added, and the missing base at the 3′ end was completed by End Repair Mix 2 in the kit. A sequencing adapter containing a library specific tag was added at the 5′ end of the sequence. BECKMAN AMPure XP Beads were used to screen and purify the library system. The DNA fragments attached to the adaptor were amplified by PCR, and the library enrichment products were purified again using BECKMAN AMPure XP Beads. The final fragment selection and purification of the library was performed by 2% agarose gel electrophoresis. The Agilent High Sensitivity DNA Kit and Quant-iT PicoGreen dsDNA Assay Kit were used to quality check the library and quantitate the library on Promega QuantiFluor system. The qualified sequencing libraries were diluted in gradient, mixed in the corresponding proportion according to the required sequencing amount, denatured with NaOH, and sequenced.

The paired-end raw reads were proceeded using Quantitative Insights Into Microbial Ecology (QIIME) pipeline [34] for quality filtering, trimming, and chimera checking. After quality checking and noise reduction, FLASH (Version 1.2.11) and Usearch (Version 10) were used to cluster the data. Operational Taxonomic units (OTUs) were clustered according to the criterion of 97% similarity, and the resulting otus represented sequences [35,36]. Finally, the Mothur method and the SSUrRNA database [37] of SILVA1323 [35] were used for species annotation analysis of OTUs sequences and the microbial community structure was counted at each taxonomic level.

*2.5. Data Processing*

One-way analysis of variance (ANOVA) and LSD ($p < 0.05$) multiple comparisons were performed on the differences in soil physicochemical properties, soil microbial community composition, and diversity under different land use patterns using SPSS [24]. The Spearman correlation coefficient method was used to analyze the correlation between soil properties and microbial community composition. The relative abundance of each soil microbial community was calculated according to different classification levels, and the microbial community composition was analyzed. The microbial community alpha diversity index was calculated according to the clustered OTUs, and based on the number of OTUs, the microbial community classification statistics were performed at each classification level through database comparison and identification, and the relative abundance of each group was calculated. RDA was used to study the correlation between soil physicochemical properties and microbial community composition. The main physicochemical, biological, and microbial diversity indicators of soil were selected, the original data were uniformly standardized, and the PLS-PM was used to study the causal relationship between the latent variables of different land use patterns and the diversity of soil microbial communities.

## 3. Results

*3.1. Analysis of Soil Physicochemical Properties under Different Land Use Patterns*

Taking the relative percentage of each particle size in the soil as the standard, the soil texture is classified into sand, silt, and clay. The particle size range of sand is 0.02–2 mm; the particle size range of the silt is 0.002–0.02 mm; the clay particle size is less than 0.002 mm [20]. In terms of soil textural composition, the soil clay content and silt content are the largest in the woodland, and the smallest in the vegetable land. The clay content is wasteland > cultivated land, and the silt content is the opposite; the order of soil sand content is vegetable land > cultivated land > wasteland > woodland, which is opposite to the soil clay content. In terms of soil pH, the pH value of vegetable land is the largest, and that of woodland is the smallest, followed by that of cultivated land and wasteland; in terms of soil total nitrogen content, total carbon content, and water content, they are the largest in cultivated land, and the smallest in wasteland, followed by vegetable land and woodland (Table 1).

**Table 1.** Soil physicochemical properties of different land use patterns in the Taojia River Basin.

| Samples | VL | WL1 | WL2 | CL | F | *p* |
|---|---|---|---|---|---|---|
| Total nitrogen (g·kg$^{-1}$) | 0.14 ± 0.03 a | 0.11 ± 0.08 a | 0.11 ± 0.01 a | 0.17 ± 0.04 a | 0.908 | 0.479 |
| Total carbon (g·kg$^{-1}$) | 16.56 ± 1.21 ab | 13.51 ± 0.45 c | 14.45 ± 1.44 bc | 17.99 ± 2.14 a | 5.929 | 0.020 |
| pH | 6.30 ± 0.29 a | 5.57 ± 0.99 a | 5.47 ± 0.14 a | 6.24 ± 0.76 a | 1.373 | 0.319 |
| moisture content (%) | 30.25 ± 4.63 a | 20.98 ± 3.02 b | 21.27 ± 2.02 b | 30.80 ± 6.04 a | 4.988 | 0.031 |
| Clay (%) | 4.92 ± 2.29 b | 10.81 ± 0.05 a | 12.23 ± 1.07 a | 9.29 ± 4.44 ab | 3.694 | 0.070 |
| Silt (%) | 30.90 ± 4.47 b | 35.33 ± 1.54 ab | 38.81 ± 0.85 a | 36.15 ± 2.37 ab | 3.439 | 0.081 |
| Sand (%) | 64.17 ± 6.71 a | 52.00 ± 0.36 b | 48.96 ± 1.93 b | 54.57 ± 6.77 ab | 4.399 | 0.049 |

Note: The data are means ± standard error (SE), different letters indicate significant levels ($p < 0.05$). VL: Vegetable land, WL1: Wasteland, WL2: Woodland, CL: Cultivated land. The same is true for the label that follows.

*3.2. Analysis of Soil Microbial Community Diversity under Different Land Use Patterns*

3.2.1. Analysis of Soil Bacterial Community Diversity

A total number of 747,083 effective sequences obtained from the 12 sequenced soil samples were optimized, filtered, and removed from low-quality sequences, resulting in an average of 62,257 valid sequences per soil sample. According to the 97% sequence similarity, the OTUs detected in vegetable land, wasteland, woodland, and cultivated land were 5124, 5149, 5137, and 5137, respectively. Among them, the total number of OTUs is 14,209, and the unique OTUs are 1292, 388, 3148, and 987, respectively. The three land use patterns of vegetable land, woodland, and cultivated land have the least number of OTUs in common, only 1. The total number of OTUs common to the two land use patterns of

woodland and cultivated land is second only to the total OTUs of soil under the four land use patterns, which is 3664. Wasteland soils have the fewest unique OTUs, only 388, which is much smaller than the OTUs, 3148, which is endemic to woodland soils (Figure A1).

In the soil of the study area, the Shannon index of bacteria in vegetable soil were the highest, 10.92, indicating that the uncertainty of OTUs occurrence of bacteria in vegetable land soil was the highest, and two sequences were randomly selected from the sample, which had the highest probability of belonging to different OTUs; the Shannon index of wasteland soil was the lowest, 10.41, and the Shannon index of cultivated soil was greater than that of woodland soil. The Simpson index of soil bacteria under the four land use patterns was very close, all of which were greater than 0.99 and close to 1.00, indicating that the dominant bacterial species had similar effects on the overall species diversity in vegetable land, woodland, cultivated land, and wasteland. The observed species index and Chao1 index of bacteria were the highest in vegetable land soil, 4077.80 and 5250.51, respectively, and the lowest in wasteland soil, 3360.93 and 4398.82, respectively, followed by cultivated land and woodland, indicating that the richness of bacterial species in vegetable soil was the highest, while the richness of bacterial species in wasteland soil was the lowest, and the richness of bacterial species in cultivated land soil was greater than that in woodland soil (Figure 2).

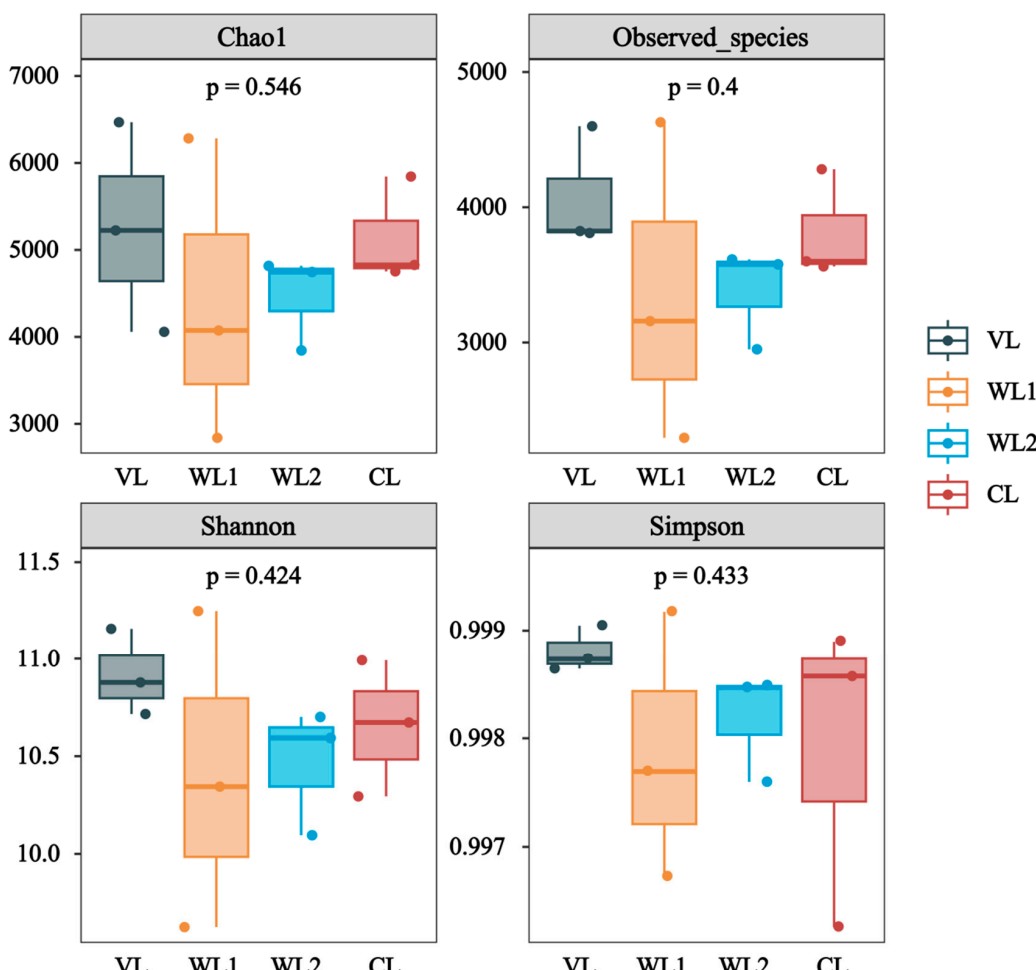

**Figure 2.** Soil bacterial indexes of different land use patterns in the Taojia River Basin.

3.2.2. Analysis of Soil Fungal Community Diversity

After optimization, and the filtration and removal of low-quality sequences, a total of 178,049 effective sequences were obtained from 12 sequenced soil samples, with an average of 14,837 valid sequences per soil sample. According to the 97% sequence similarity, the

OTUs detected in vegetable land, wasteland, woodland, and cultivated land soil were 5577, 8762, 8414, and 3093, respectively. Among them, the total number of OTUs is 9615, and the unique OTUs are 989, 2566, 1783, and 3, respectively. The number of common OTUs in soil under the four land use patterns was 2767. The three land use patterns of vegetable land, woodland, and cultivated land have no number of OTUs in common, and the common OTUs number of wasteland, woodland, and cultivated land is also 0. Meanwhile, there is no common OTU number between cultivated land and vegetable land or between cultivated land and woodland. Woodland soils have the fewest endemic OTUs, only 3, which is much smaller than the endemic OTUs in wasteland land soils: 2566 (Figure A2).

In the soils of the study area, the Shannon index, observed species index, and Chao1 index of fungal species in woodland soil were the highest, which were 3.95, 186.67, and 184.25, respectively, and the lowest in cultivated land soil, at 3.08, 137.33, and 139.58, respectively, followed by wasteland and vegetable land. The Simpson index of wasteland soil was the lowest, 0.81; the Simpson index of woodland soil was smaller than that of vegetable land soil; the Simpson index of cultivated land soil was the highest, 0.96, showed that the dominant species had the greatest impact on the overall species diversity in cultivated land, with species diversity being relatively low. However, a few dominant species in the wasteland had the least effect on the overall species diversity, which was relatively high. The results indicating that the species richness of fungi in woodland soil was the highest, the species richness of fungi in cultivated land soil were the lowest, followed by wasteland and vegetable land. (Figure 3)

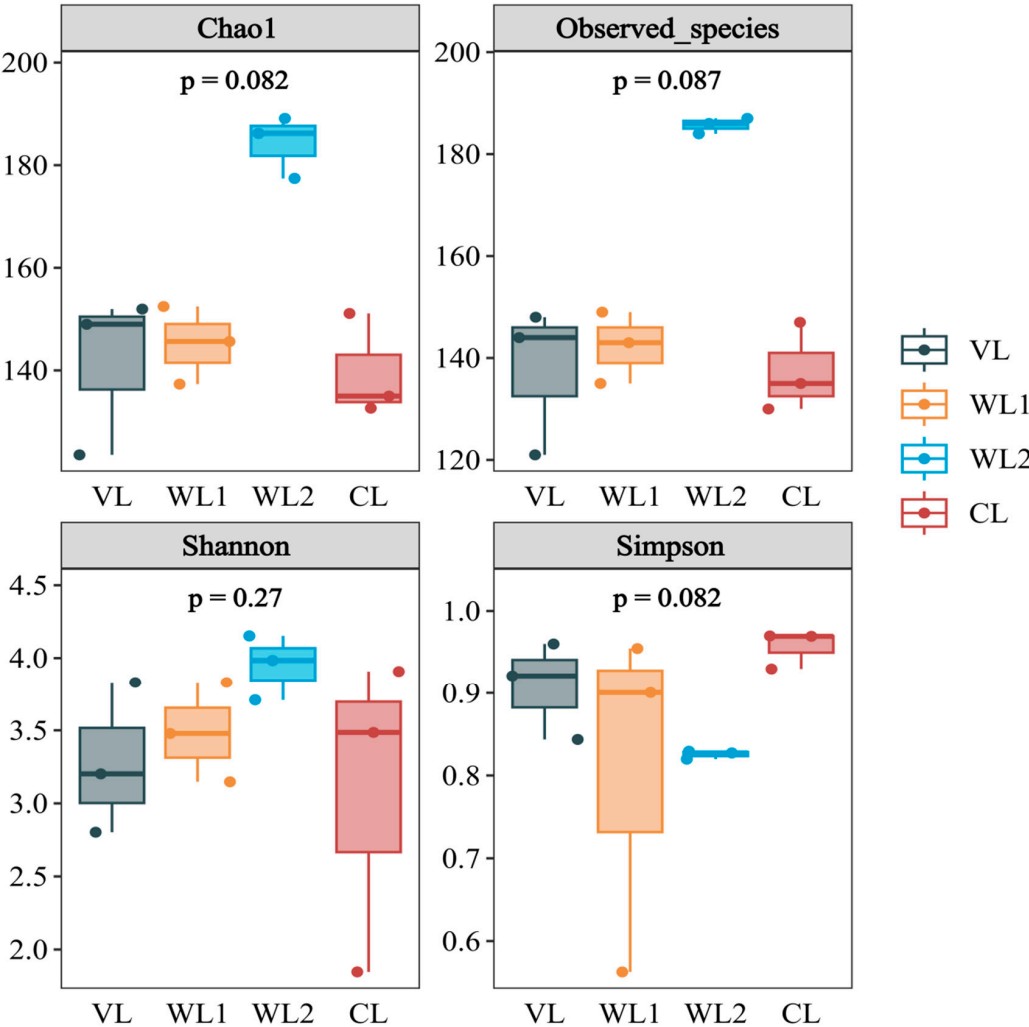

**Figure 3.** Soil fungus indexes of different land use patterns in the Taojia River Basin.

### 3.3. Analysis of Soil Microbial Community Composition Characteristics under Different Land Use Patterns

3.3.1. Analysis of Soil Bacterial Community Composition

Through sequencing data analysis, the bacteria in 12 soil samples from the four different land use patterns of vegetable land, wasteland, woodland, and cultivated land could be divided into 48 phyla, 134 classes, 1037 families, 536 genera, and 1037 species. The community composition showed that at the phylum level, the dominant phylum of vegetable land, wasteland, woodland, and cultivated land was *Proteobacteria*, with a relative abundance of 20.69%~32.70%. The phyla with an average relative abundance of more than 10% were *Actinobacteria* (15.78%~39.69%), *Chloroflexi* (17.43%~21.48%), and *Acidobacteria* (12.11%~15.90%). The remaining phyla with an average relative abundance of more than 5% were *Gemmatimonadetes* (2.93%~5.17%) and *Firmicutes* (0.63%~13.18%) (Figure 4). There were significant differences in the relative abundance of *Proteobacteria* ($p = 0.039$), *Nitrospirae* ($p = 0.041$), *Latescibacteria* ($p = 0.035$), and *Planctomycetes* ($p = 0.032$) among different land use patterns ($p < 0.05$), while the relative abundances of bacteria in the remaining phyla showed no significant difference ($p > 0.05$). The differences in the relative abundance of bacteria in specific phyla are shown in Table A1.

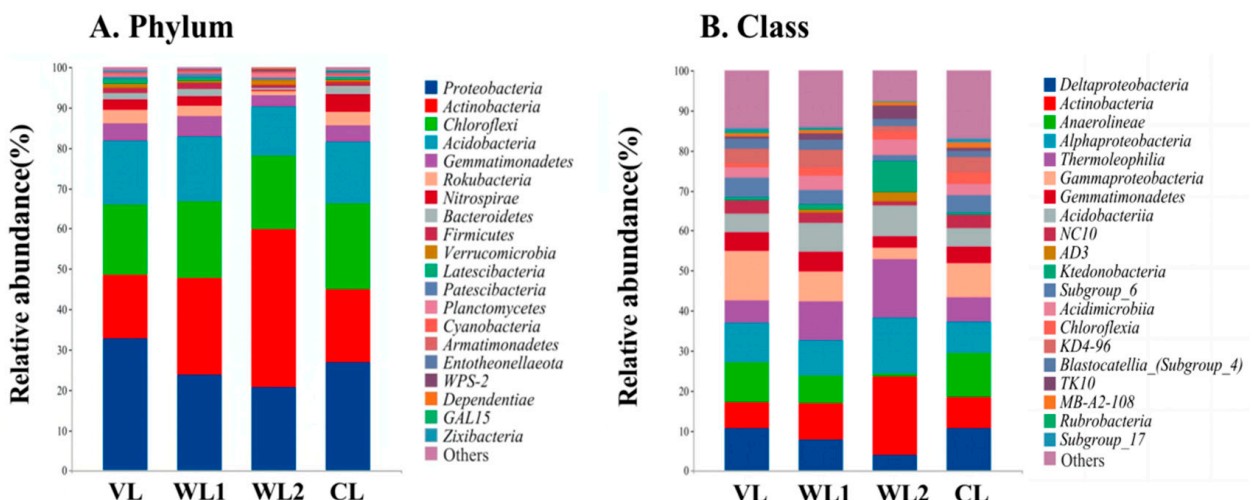

**Figure 4.** The relative abundances of bacterial communities in the Taojia River Basin. Note: (**A**) is phylum level, (**B**) is Class level.

At the class level, the main bacterial groups with an average relative abundance greater than 10% in vegetable land, wasteland, woodland, and cultivated land were *Actinobacteria* (7.99%~16.95%). Other genera and taxa with an average relative abundance of more than 5% were *Alphaproteobacteria* (7.68%~13.86%), *Thermoleophilia* (5.89%~14.64%), *Deltaproteobacteria* (3.99%~10.68%), *Gammaproteobacteria* (2.83%~12.35%), *Anaerolineae* (0.89%~11.26%), *Acidobacteria* (4.63%~7.58%), *Gemmatimonadetes* (2.89%~5.08%), Subgroup_6 (1.73%~5.02%), and *Ktedonobacteria* (0.41%~7.53%) (Figure 4). There were significant differences in the relative abundance of *Deltaproteobacteria* ($p = 0.043$), *Actinobacteria* ($p = 0.04$), *Alphaproteobacteria* ($p = 0.004$), *Gammaproteobacteria* ($p = 0.034$), *TK10* ($p = 0.013$), and *Subgroup_17* ($p = 0.04$) among different land use patterns ($p < 0.05$). The differences in the relative abundance of specific class bacteria are shown in Table A2.

3.3.2. Correlation Analysis of Soil Bacterial Community Composition and Soil Physicochemical Properties

The results of Spearman analysis showed that the two dominant phylum with relative abundances of more than 20% showed a certain correlation with soil factors at the bacterial phylum level. *Actinobacteria* showed a significant correlation with pH value and sand content of 0.01, which was negatively correlated and positively correlated with clay content

(*p* < 0.01). *Proteobacteria* and *Rokubacteria* were positively correlated with pH value and sand content (*p* < 0.01), and negatively correlated with clay content (*p* < 0.01). *Latescibacteria* were significantly negatively correlated with silt content (*p* < 0.01) (Figure A3).

At the level of bacterial classes, the dominant orders *Actinomycetes*, *Thermoleophilia*, *Deltaproteobacteria*, *Subgroup_6*, *Acidobacteriia*, *NC10*, and *TK10* were significantly correlated with pH (*p* < 0.01), and were significantly negatively correlated with *Actinomycetes*, *Thermoleophilia*, *Acidobacteriia*, and *TK10*, and positively correlated with the other two taxa (*p* < 0.01). *Deltaproteobacteria*, *Subgroup_6*, and *NC10* were positively correlated with sand content and negatively correlated with clay content (*p* < 0.01). *Actinomycetes*, *Thermoleophilia*, *Acidobacteriia* and *TK10* were negatively correlated with sand grain content and clay content (*p* < 0.01) (Figure 5).

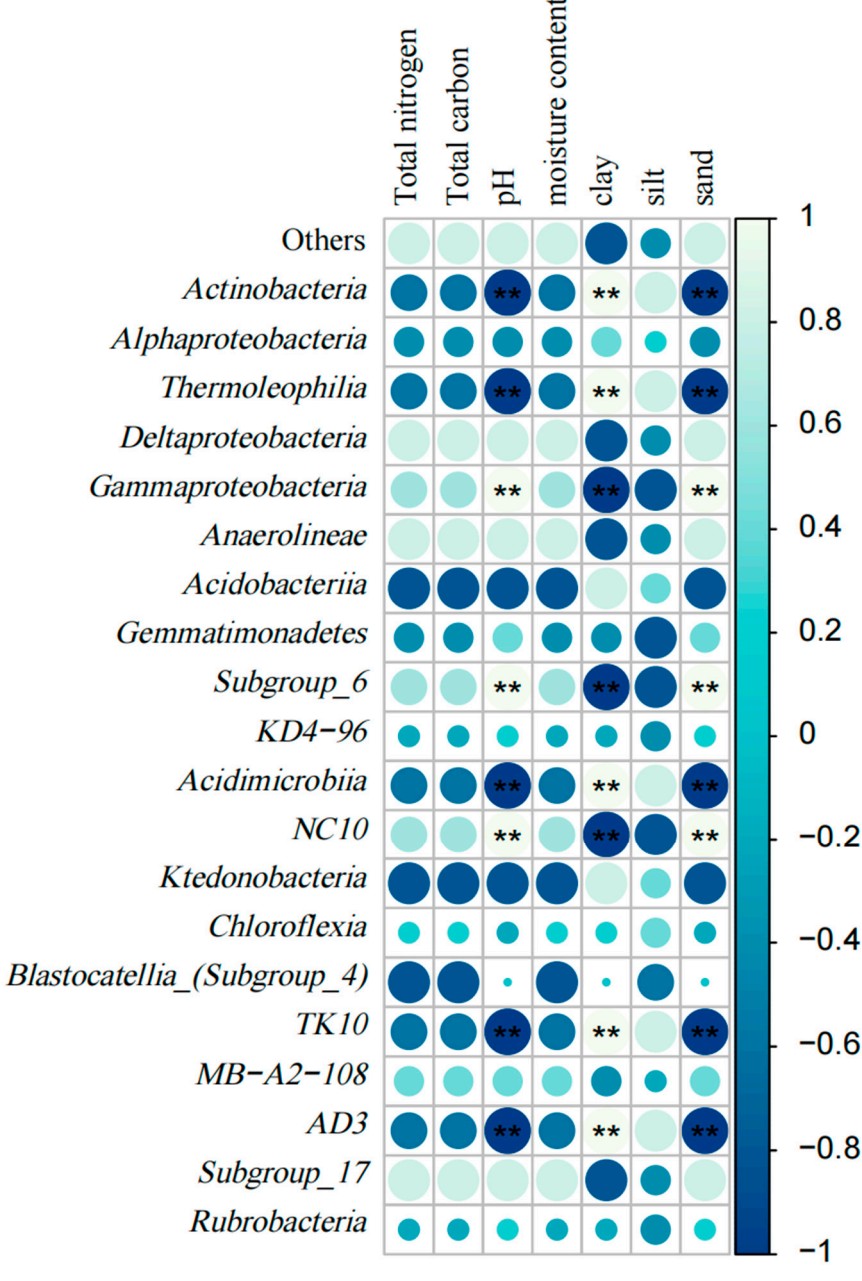

**Figure 5.** Spearman's rank correlation analysis of soil factor and bacterial community at class level in the Taojia River Basin. Note: The size of circle indicates the strength of correlation, ** means *p* < 0.01.

RDA was used to explore the relationship between soil bacterial community and soil properties under four land use patterns. This analysis (Figure 6) showed that RDA1 and RDA2 explained 15.95% and 26.19% of soil bacterial community composition, respectively. Soil clay and pH had a greater effect on bacterial community composition, soil TC had the least effect on bacterial community composition. Among them, clay, pH, sand, and moisture content had significant effects on the structural changes of bacterial communities. pH was positively correlated with bacterial communities in vegetable land, and TC, TN, moisture content, and sand were positively correlated with bacterial communities in cultivated land. In addition, the dominant bacterial phylum (relative abundance >1%, top 10) was associated with soil properties such as clay, sand, silt, moisture content, pH, TC, and TN. For example, the abundance of *Actinobacteria* is positively correlated with soil clay and silt, and negatively correlated with soil sand, moisture content, TC, and TN. The abundance of *Proteobacteria*, *Nitrospirae*, *Bacteroidetes*, and *Zixibacteria* was positively correlated with soil sand, moisture content, TC, and TN. The abundance of *Patescibacteria* is inversely correlated with pH.

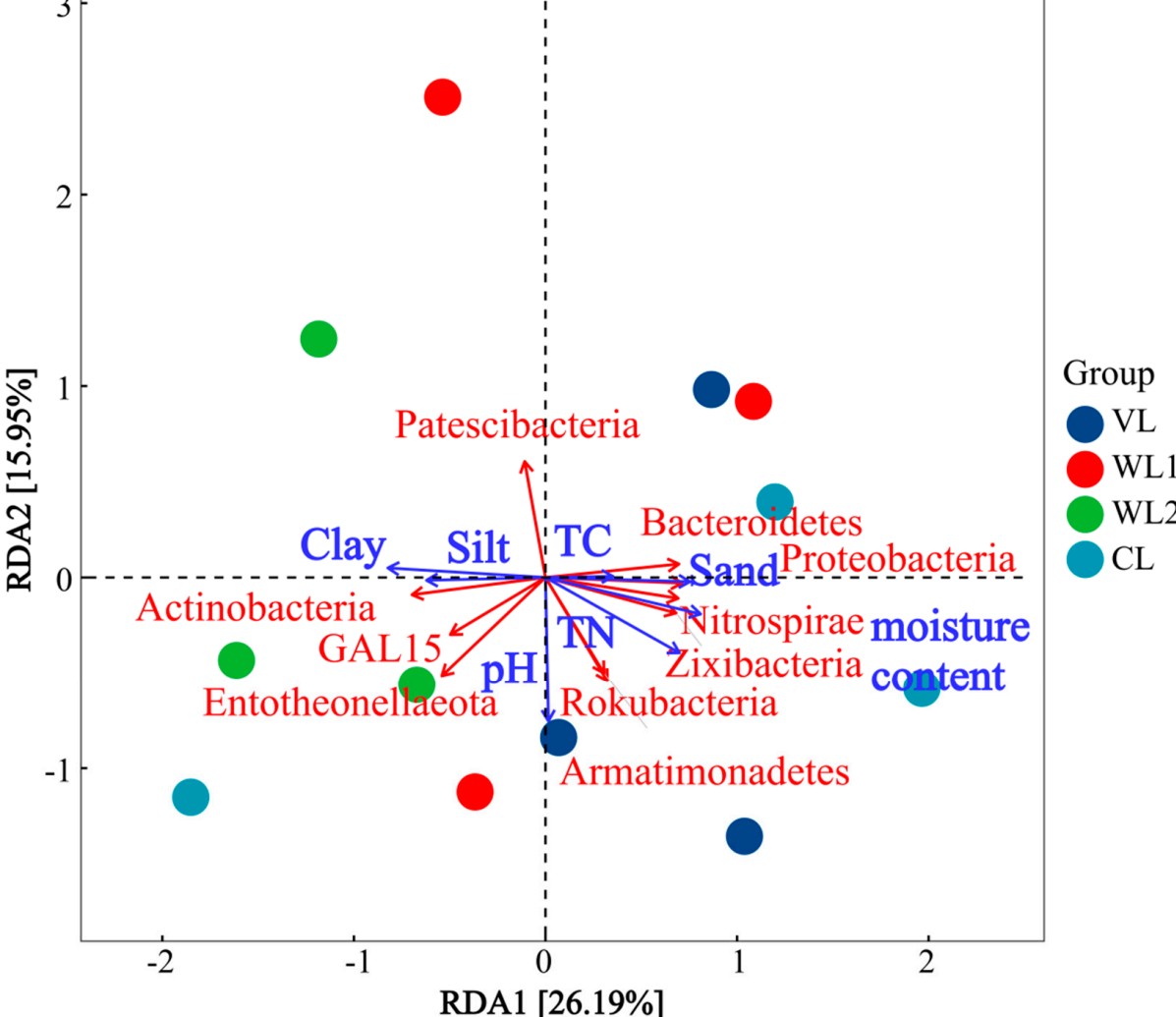

**Figure 6.** RDA of bacterial communities under different land use patterns in the Taojia River Basin. Note: The arrows (blue−effects of soil properties on bacterial communities; red−effects on the dominant bacterial phyla) indicate the lengths and angles between explanatory and response variables and reflect their correlations. TN (total nitrogen), TC (total carbon).

### 3.3.3. Analysis of Soil Fungal Community Composition

Through sequencing data analysis, the microorganisms in a total of 12 soil samples in the four different land use patterns of vegetable land, wasteland, woodland, and cultivated land can be divided into 7 phyla, 40 classes, 138 families, 155 genera, and 113 species. At the phylum level, the dominant phyla of vegetable land, woodland, and cultivated land were *Mucoromycota*, 29.39%, 41.36%, and 22.67%, respectively. *Ascomycota* is the most abundant in wasteland, at 42.16%. Phyla with an average relative abundance of more than 10% include *Basidiomycota* (13.18%~22.28%) and *Chytridiomycota* (1.01%~14.76%) (Figure 7). The level of fungal phyla did not differ significantly across land use patterns ($p > 0.05$). The differences in fungal abundance of specific phyla are shown in Table A3. The dominant phylum *Mucoromycota* had the greatest difference in abundance between woodland and wasteland, *Ascomycota* had the greatest difference in abundance between wasteland and vegetable land.

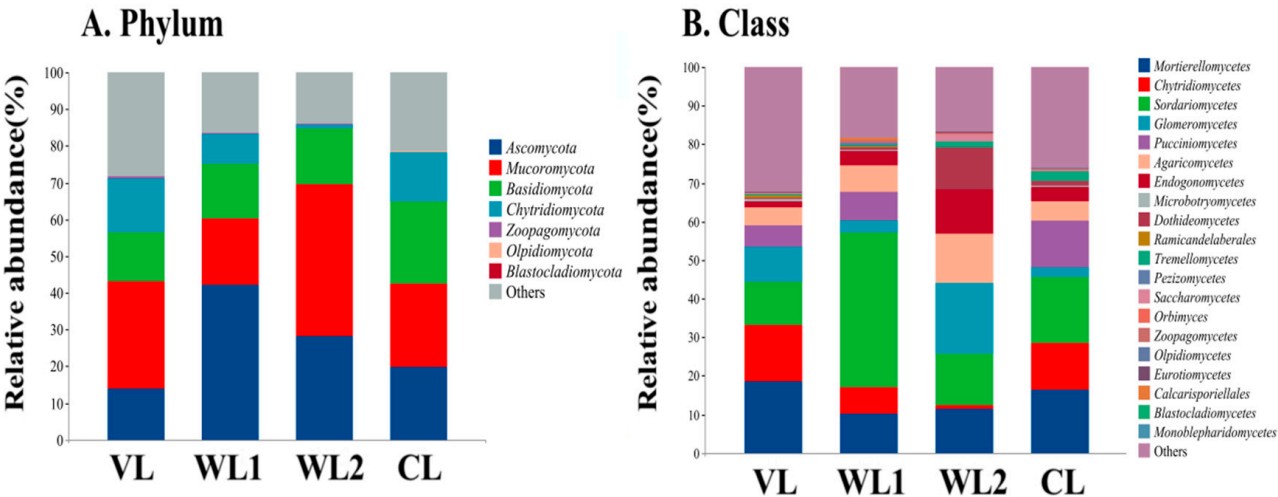

**Figure 7.** The relative abundances of fungus communities in the Taojia River Basin. Note: (**A**) is phylum level, (**B**) is Class level.

At the class level, the main fungal taxa with a relative abundance of greater than 10% that were detected in vegetable land, wasteland, woodland, and cultivated land were *Sordariomycetes* (11.02%~39.99%) and *Mortierellomycetes* (10.27%~18.73%). *Sordariomycetes* had the largest relative abundance in wasteland and cultivated land: 18.73% and 16.92%, respectively. *Mortierellomycetes* is the most abundant in vegetable land, at 24.88%; *Glomeromycetes* had the largest relative abundance in woodland, which was 18.38% (Figure 7). *Blastocladiomycetes* (0.18%) were found only in vegetable land. Among the woodlands, *Glomeromycetes* and *Agaricomycete*s showed the highest relative abundance, which was significantly different from the other three land use patterns ($p < 0.05$). The differences in fungal abundance in specific phyla are shown in Table A4.

### 3.3.4. Correlation Analysis of Soil Fungal Community Composition and Soil Physicochemical Properties

The results of Spearman analysis showed that *Chytridiomycota* had significant negative correlations with clay ($p < 0.01$), which was significantly positively correlated with pH and sand ($p < 0.01$) (Figure A4).

At the fungal class level, *Microbotryomycetes* and *Chytridiomycetes* showed significant negative correlations with clay ($p < 0.01$), and positively correlated with pH and sand ($p < 0.01$). *Agaricomycetes* and *Endogonomycetes*, on the contrary, were significantly positively correlated with clay ($p < 0.01$), and negatively correlated with pH and sand ($p < 0.01$). *Dothideomycetes* were significantly positively correlated with silt content ($p < 0.01$), while

*Zoopagomycete*s were significantly negatively correlated with soil TC, TN, and moisture content ($p < 0.01$) (Figure 8).

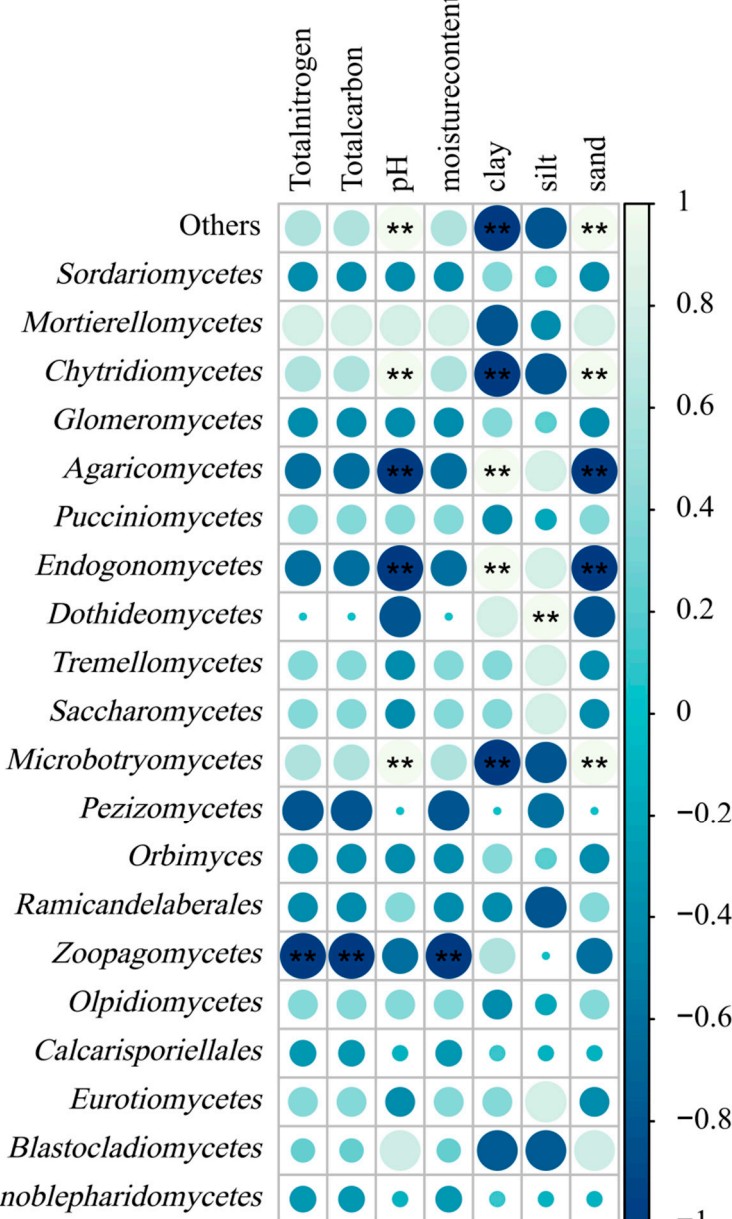

**Figure 8.** Spearman's rank correlation analysis of soil factor and fungus community at class level in the Taojia River Basin. Note: The size of circle indicates the strength of correlation, ** means $p < 0.01$.

RDA was used to explore the relationship between soil fungal community composition and soil physicochemical properties under four land use patterns. This analysis showed (Figure 9) that RDA1 and RDA2 explained 31.04% and 25.61% of soil fungal community composition, respectively. The fungal community composition of woodland soil was more similar than that of other land use patterns. TN was positively correlated with moisture content, there was a negative correlation between pH and moisture content. Soil clay and moisture content had a greater effect on fungal community composition, silt had the least effect on fungal community composition. Among them, pH was positively correlated with fungal community composition in wasteland and negatively correlated with fungal community composition in vegetable land, cultivated land, and woodland. Soil TN and moisture content were positively correlated with soil fungal community composition in

vegetable land, cultivated land, and woodland, but negatively correlated with soil fungal community composition in wasteland. The dominant fungal phylum was associated with soil properties such as clay, pH, silt, moisture content, and TN. For example, the abundance of *Ascomycota* in wasteland increased with increasing soil pH and increased with decreasing soil silt and clay. With the decreases of soil TN and moisture content, the relative abundance of *Mucoromycota* and *Basidiomycota* decreased in the four land use patterns.

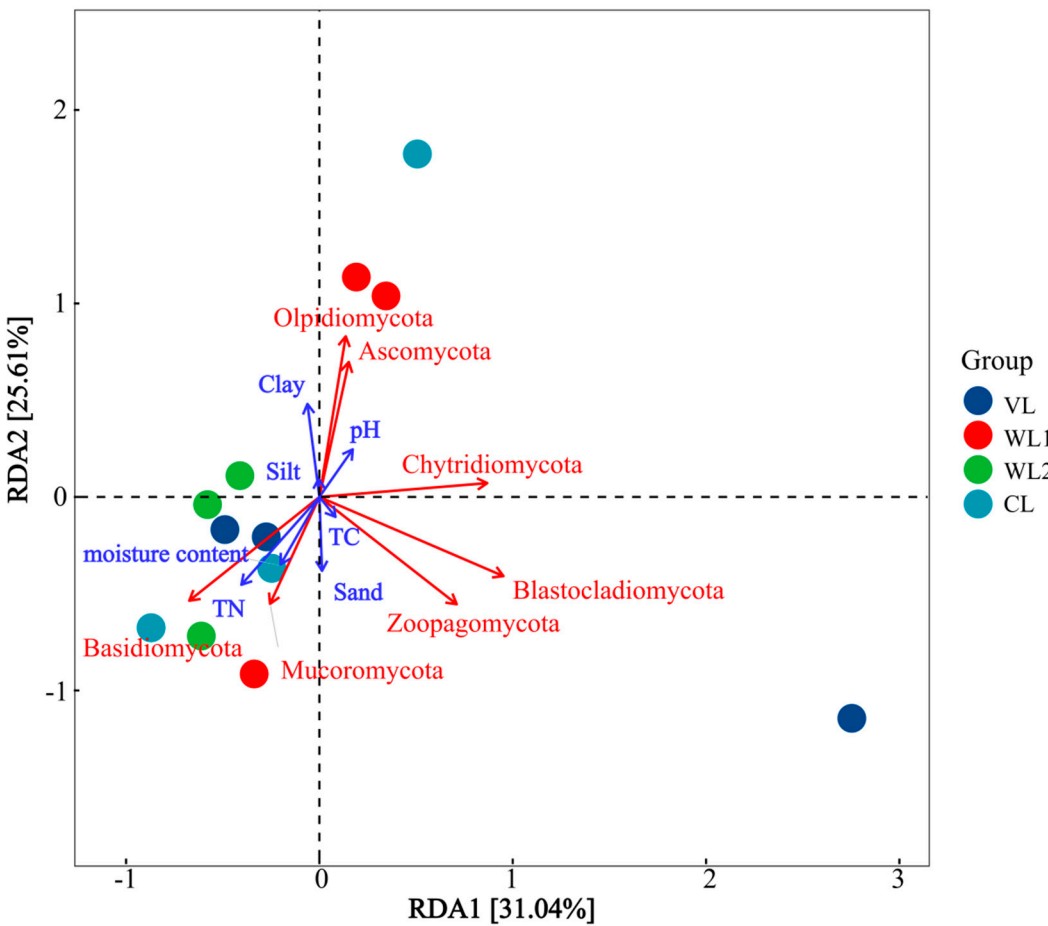

**Figure 9.** RDA of fungus communities under different land use patterns in the Taojia River Basin. Note: The arrows (blue—effects of soil properties on fungal communities; red—effects on fungus phyla) indicate the lengths and angles between explanatory and response variables and reflect their correlations. TN (total nitrogen), TC (total carbon).

### *3.4. PLS-PM of Soil Bacterial and Fungal Community Diversity*

Using R language, the partial least-squares path model was established by optimizing the prediction of dependent variables and fitting the data to the predetermined model and expressing the causal relationship between latent variables through linear conditional expectation. According to the prediction results and conjectures, clay, sand, and silt were artificially classified as soil textural composition, soil moisture content as soil humidity, and soil pH, TC, and TN as soil chemical properties. The goodness-of-fit (GOF) of the model was 0.737, indicating that the overall explanatory ability of the model was a good measure. It can be seen that land use pattern has a positive effect on soil textural composition and soil humidity and has a very significant impact on soil humidity ($p < 0.01$) and a negative impact on soil chemical properties. Soil textural composition has a significant positive effect on soil chemical properties ($p < 0.01$) and a very significant negative effect on soil humidity ($p < 0.001$), while soil humidity has an extremely significant positive effect on soil chemical properties ($p < 0.001$). Soil textural composition, soil moisture, and

soil chemical properties all had effects on bacterial and fungal community diversity, but none of them was significant. The explanatory degree ($R^2$) of soil bacteria and soil fungi was 0.26 and 0.73, respectively, indicating that fungi were more affected by different soil physicochemical factors than bacteria, that is, different land use patterns had greater effects on soil fungal community diversity than on soil bacterial community diversity. Effect represents the influence state of one variable on another variable, soil humidity and soil chemical properties have a positive total effect on soil bacteria and fungal community diversity, soil textural composition has negative effects on fungal and bacterial community diversity. Land use pattern has a positive total effect on soil bacterial community diversity and has a negative total effect on soil fungal community diversity, soil bacterial community diversity has effect on soil fungal community diversity (Figure 10).

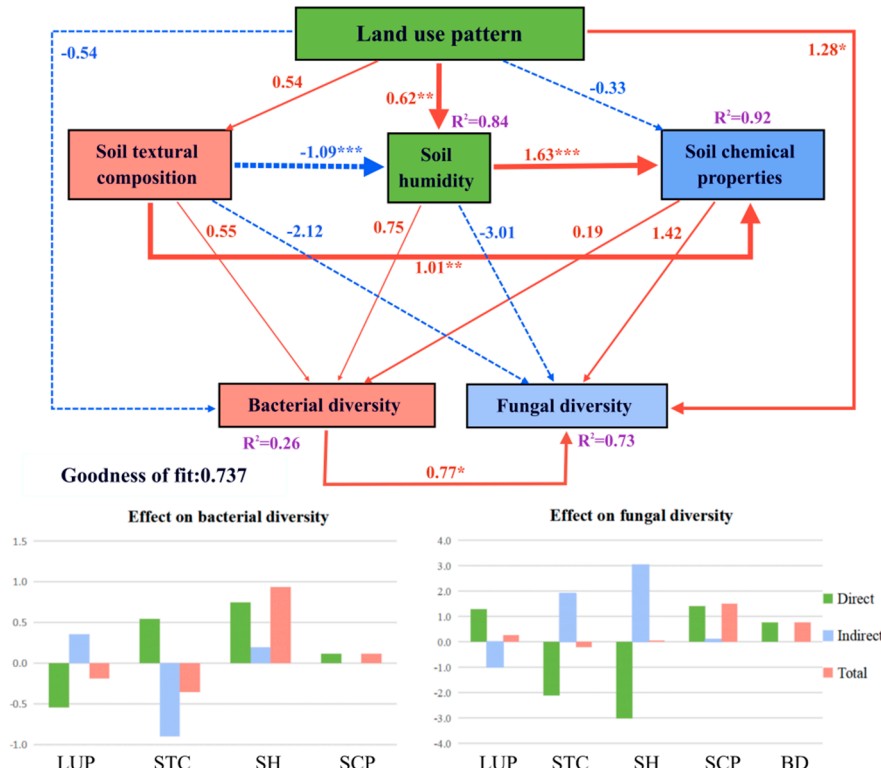

**Figure 10.** The PLS−PM of soil bacterial and fungal communities' diversity in the Taojia River Basin. Note: Red represents positive path, blue represents negative path, solid line represents significant path, dotted line represents insignificant path. The number on the arrow represents the path coefficient. The width of the arrow indicates the size of the significance. * means $p < 0.05$, ** means $p < 0.01$, *** means $p < 0.001$. $R^2$ represents the proportion of the factor explained, LUP represents Land use pattern, STC represents soil textural composition, SH represents soil humidity, SCP represents soil chemical properties, BD represents bacterial diversity.

## 4. Discussion

Land use change can directly inference the physicochemical properties and structure of soil, thus affecting the difference of soil microbial community composition [38]. In this study, it was found that the dominant phylum of bacteria in vegetable land, wasteland, woodland, and cultivated land was *Proteobacteria*, with a relative abundance of 20.69% to 32.70%. This result is consistent with Yang Wen et al., who found that *Proteobacteria* were the dominant bacteria in the soil after succession of old fields in the Loess Plateau of China [39]. *Proteobacteria* are mostly *facultative* or aerobic bacteria, which are widely distributed and have strong adaptability to the environment. They have high abundance in the soil of different land use patterns in the study area. This study found that the dominant bacteria in cultivated land include *Firmicutes*, *Alphaproteobacteria*, and *Gammaproteobacteria*.

The results were similar to those of Eiko E. Kuramae et al. on the soil bacterial community of the six most important land use types in the Netherlands [40]. The relative abundance of the dominant phylum *Proteobacteria* and the dominant class *Actinobacteria* were significant differences under different land use patterns. *Deltaproteobacteria*, *Alphaproteobacteria*, and *Gammaproteobacteria* belonging to the *Proteobacteria* were significantly different among different land use patterns, echoed the above conclusion. In vegetable land, woodland, and cultivated land, the main dominant fungi phyla were *Mucoromycota*, the relative abundance of *Ascomycota* was highest in wasteland. The dominant phylum *Mucoromycota* had the greatest difference in abundance between woodland and wasteland, *Ascomycota* had the greatest difference in abundance between wasteland and vegetable land, which is different from Jiyi et al.'s results, which indicate that the dominant phyla of soil fungi in the five land use types in the karst area are *Ascomycota*, *Basidiomycota*, and *Mortierellomycota* [41]. This may be because different fungi differ in their ability to efficiently decompose organic matter in soil. The results of Spearman analysis showed that the dominant groups in the bacterial and fungal communities had significant correlations with soil pH, clay, and sand. These results indicated that soil texture composition has an important effect on soil microbial community composition in river basin. The RDA results showed that soil clay, pH, and moisture were the key environmental factors affecting soil microbial communities. It is inconsistent with previous research results that soil TN, TC, and other components play a key role in soil microbial community composition [42,43]. The soil samples contained multiple classes and groups of *Firmicutes* (*Bacilli*, *Clostridia*, *Erysipelotrichia*, *Negativicutes*) and multiple genera and groups of *Bacteroidetes* (*BSV26*, *SJA-28*, *Flavobacterium*, etc.), based on previous researchers who found that the relative abundance of *Firmicutes* and *Bacteroidetes* was higher in areas with higher soil heavy metal content [44,45]. Therefore, whether the soil in the study area is affected by heavy metals needs further study.

Different land use patterns not only affect the difference of microbial community structure, but also affect the diversity of soil microbial community [38]. In this study, the effect of land use patterns on the diversity of soil microbial communities showed significant differences under the four different land use patterns. The results showed that the bacterial diversity of vegetable land was higher than that of other land use patterns, and the fungal diversity of woodland was also significantly higher than that of other land use patterns. The diversity indices of bacteria and fungi varied between different land use patterns, and fungi were more affected by different land use patterns than bacteria. The result is different from the study by Jiao et al., where the bacterial and fungal community diversity in the soil of woodland in the Loess Plateau was significantly higher than that in farmland [46]. The higher bacterial diversity in vegetable plots compared to other land use types may be due to long-term management practices such as tillage and organic fertilizer application, which have a certain impact on the soil microbial community, promoting or inhibiting the growth of certain microorganisms, thus enhancing bacterial diversity, which can demonstrate Dong Xiongde et al.'s results, which found that the application of chemical fertilizers can increase the potential microbial diversity [26]. Soil textural composition can affect microbial diversity by affecting soil microenvironment, including aeration, moisture, nutrients, and aggregates [47]; this study found that soil textural composition has a very important positive effect on soil chemical properties, while they have a significant negative impact on soil moisture content, Meanwhile, soil moisture content has a significant positive impact on soil chemical properties. Soil chemical properties include pH, TN, and TC. Soil pH directly or indirectly affects the diversity of soil microorganisms [48]. In this study, it was found that soil TC and TN have a smaller impact on soil diversity, which is different from previous studies that found soil nutrients have a greater impact on the diversity and abundance of soil microbial communities [49,50]. The lowest diversity of bacteria in barren land may be due to the lack of nutrients and organic matter in the soil, lack of coverage, long-term exposure of the soil surface to sunlight, and extreme dryness of the soil, which hinder the survival and reproduction of bacteria. The lowest fungal diversity in cultivated land may be influenced by humidity, pH, and TN, while seasonal factors also need to be

considered, which is a conclusion similar to that of Ji Chuning et al. that pH value and water content have significant effects on soil fungal community diversity in the study of fungal community succession of reclaimed soil in mining areas [51]. Many factors, such as seasonal variation, land use, and soil management have a significant impact on bacterial richness and diversity [52,53]. The sampling time is in the spring, and the spring is the natural growth of plants, different vegetation differences, will make the physicochemical properties of soil changes, thus affecting soil microorganisms. Human's agricultural management measures to different areas in spring and the change of climate environment in spring may change the soil characteristics of the sample land, and then affect soil microorganisms. Soil texture composition, soil moisture, and soil chemical properties all have an impact on the diversity of soil fungi and bacterial communities. Aside from soil texture composition negatively impacting soil microbial community diversity, other soil physicochemical properties overall have a positive impact on soil microbial community diversity. Therefore, it can be seen from the above results that soil physicochemical properties are the main factors that affect microbial diversity and community structure [54,55].

## 5. Conclusions

In this study, high-throughput sequencing technology was used to study the soil microbial analysis and diversity community composition of four different land use patterns (vegetable land, wasteland, woodland, and cultivated land) in Madi Village, Fangyuan Town, Guiyang County, Hunan Province. The results of Spearman analysis showed that the dominant groups in the bacterial and fungal communities had significant correlations with soil pH, clay, and sand. These results indicated that soil texture composition has an important effect on soil microbial community composition in river basin. The RDA results showed that soil clay, pH, and moisture were the key environmental factors affecting soil microbial communities. The relative abundance of *Proteobacteria* and *Actinobacteria* was the highest in the bacterial communities and *Mucoromycota* and *Ascomycota* was the highest in the fungi. Through the analysis of alpha diversity, it was known that bacteria and fungi had the highest species richness in vegetable land and woodland, respectively. The construction of PLS-PM demonstrated that soil fungi were more affected by land use patterns than soil bacteria in the Taojia River Basin. Based on the above results, it is speculated that woodland fungal communities in river regions may play an important role in the whole ecosystem. Therefore, forest management and protection are the key to soil ecological restoration in the Taojia River Basin. In summary, different land use patterns in the Taojia River Basin determine all aspects of soil microorganisms, including soil physicochemical properties, microbial diversity and community composition, and the correlation of soil environmental factors. In most of the previous studies, the effects of soil texture composition on soil microbial community composition and diversity were not significant. Therefore, this study has important theoretical and practical significance for revealing more internal relationships between land use patterns and soil microbial diversity and community composition, analyzing soil microbial environment, and maintaining ecosystem stability.

**Author Contributions:** Z.H.: conceptualization, methodology, formal analysis, investigation, writing—original draft, writing—review and editing, visualization, project administration. C.Y.: conceptualization, investigation, writing—review and editing, supervision, project administration. P.C.: formal analysis, writing—review and editing, visualization, investigation. Z.R.: validation, writing—review and editing. T.P.: validation, writing—review and editing. T.H.F.: formal analysis, writing—review and editing. G.W.: conceptualization, writing—review and editing, methodology, formal analysis. W.Y.: resources, writing—review and editing, supervision, project administration, funding acquisition. J.W.: conceptualization, writing—review and editing, supervision, project administration, funding acquisition. All authors have read and agreed to the published version of the manuscript.

**Funding:** This study was supported by the National Natural Science Foundation of China (42007383), Special Projects for the Construction of Chenzhou National Sustainable Development Agenda Innovation Demonstration Zone (2019sfq36).

**Data Availability Statement:** The data presented in this study are available on request from the corresponding author. The data are not publicly available due to the funded projects have not been completed.

**Acknowledgments:** Thanks are also given to the staff of Lutou and Nanshan National Station for Scientific Observation and Research of Forest Ecosystems for field sampling and laboratory analysis.

**Conflicts of Interest:** The authors declare no conflict of interest.

## Appendix A

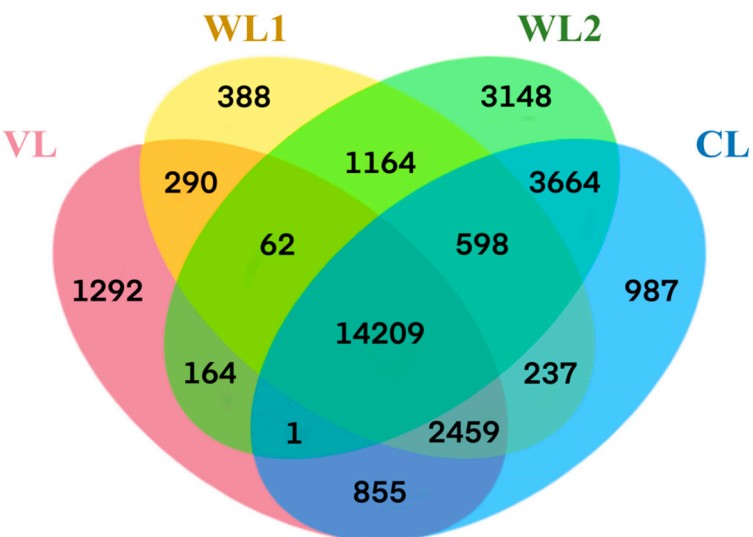

**Figure A1.** Venn diagram of Bacterial community OTUs observed under different land use patterns in the Taojia River Basin.

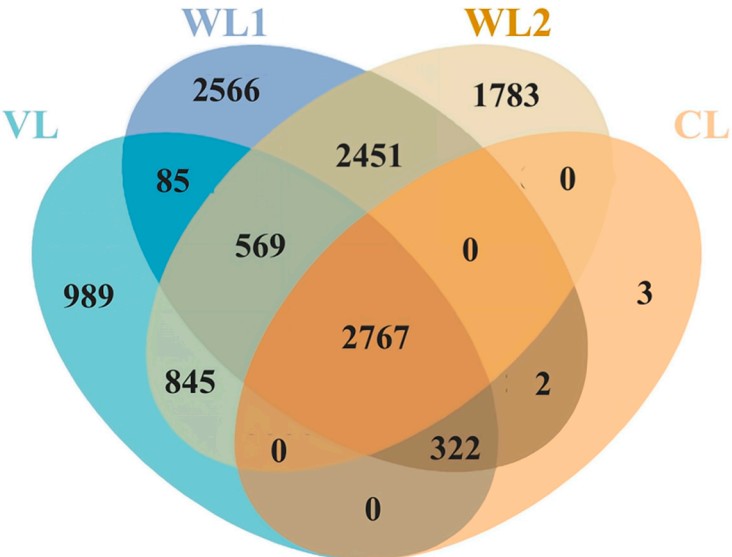

**Figure A2.** Venn diagram of Fungal community OTUs observed under different land use patterns in the Taojia River Basin.

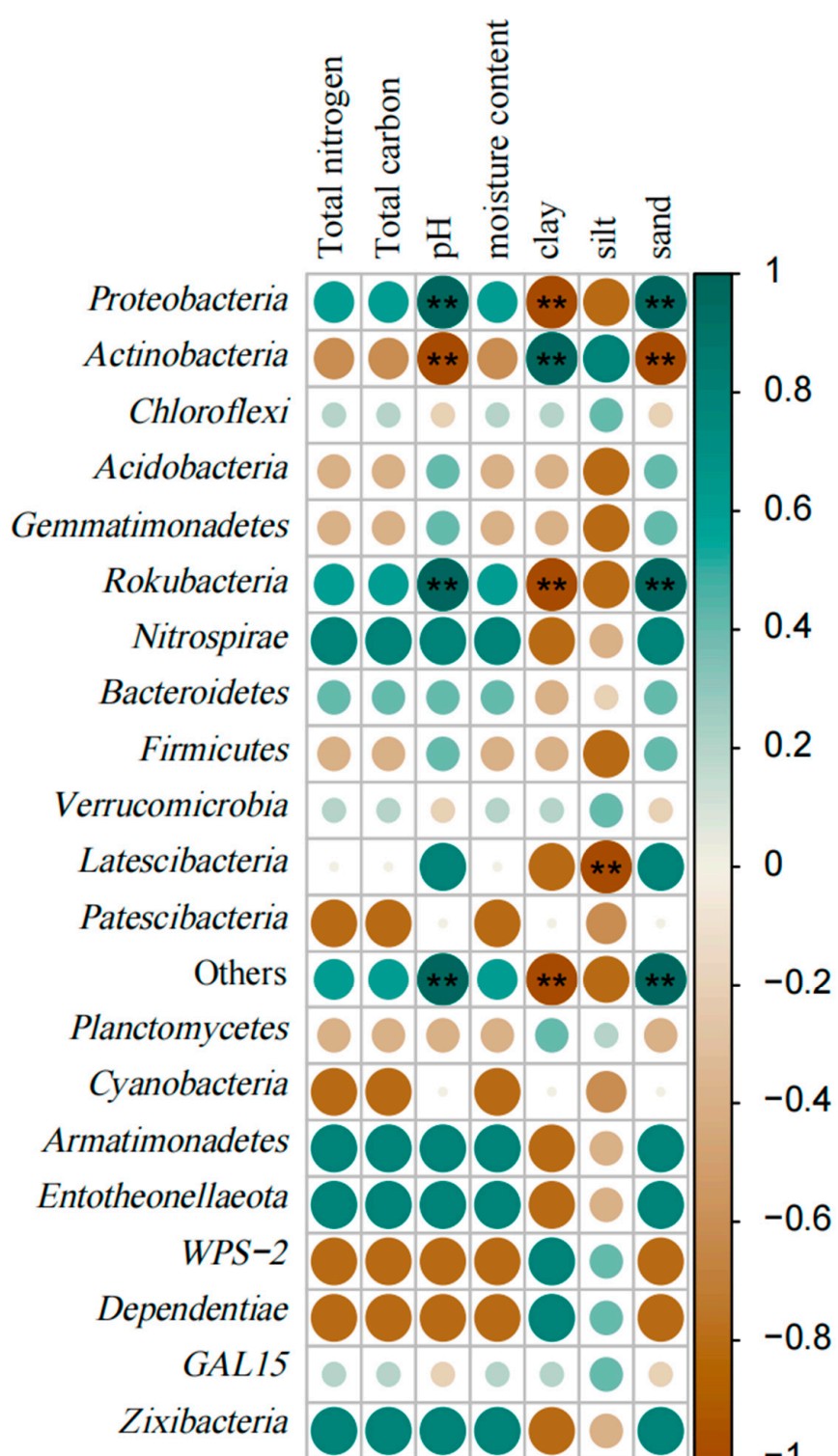

**Figure A3.** Spearman's rank correlation analysis of soil factor and bacterial community at phylum level in the Taojia River Basin. Note: The size of circle indicates the strength of correlation, ** means *p* < 0.01.

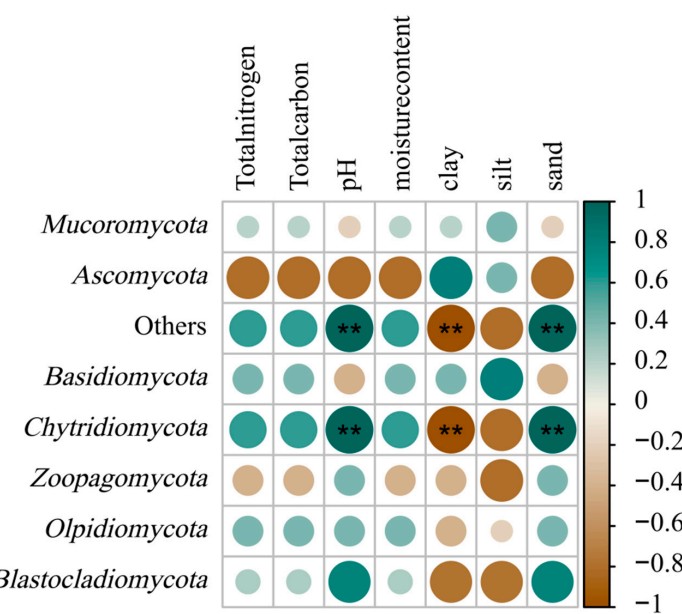

**Figure A4.** Spearman's rank correlation analysis of soil factor and fungus community at phylum level in the Taojia River Basin. Note: The size of circle indicates the strength of correlation, ** means $p < 0.01$.

**Table A1.** Comparison of relative abundance of bacterial phylum with significant differences among different samples in the Taojia River Basin.

| Taxon | VL | WL1 | WL2 | CL |
|---|---|---|---|---|
| Proteobacteria | $0.33 \pm 0.01$ a | $0.24 \pm 0.01$ b | $0.21 \pm 0.03$ b | $0.27 \pm 0.08$ ab |
| Chloroflexi | $0.17 \pm 0.08$ a | $0.19 \pm 0.09$ a | $0.18 \pm 0.11$ a | $0.21 \pm 0.03$ a |
| Actinobacteria | $0.16 \pm 0.09$ a | $0.24 \pm 0.10$ a | $0.39 \pm 0.15$ a | $0.18 \pm 0.18$ a |
| Acidobacteria | $0.16 \pm 0.01$ a | $0.16 \pm 0.03$ a | $0.12 \pm 0.03$ a | $0.15 \pm 0.03$ a |
| Gemmatimonadetes | $0.05 \pm 0.01$ a | $0.05 \pm 0.01$ a | $0.03 \pm 0.01$ a | $0.04 \pm 0.01$ a |
| Rokubacteria | $0.03 \pm 0.01$ a | $0.03 \pm 0.02$ a | $0.01 \pm 0.01$ a | $0.03 \pm 0.00$ a |
| Nitrospirae | $0.02 \pm 0.01$ ab | $0.02 \pm 0.02$ ab | $0.00 \pm 0.00$ b | $0.05 \pm 0.03$ a |
| Firmicutes | $0.01 \pm 0.01$ a | $0.02 \pm 0.01$ a | $0.01 \pm 0.00$ a | $0.01 \pm 0.01$ a |
| Bacteroidetes | $0.02 \pm 0.01$ a | $0.02 \pm 0.01$ a | $0.01 \pm 0.00$ a | $0.02 \pm 0.01$ a |
| Latescibacteria | $0.01 \pm 0.01$ a | $0.01 \pm 0.00$ ab | $0.00 \pm 0.00$ b | $0.01 \pm 0.00$ ab |
| Verrucomicrobia | $0.01 \pm 0.01$ a | $0.00 \pm 0.00$ a | $0.01 \pm 0.00$ a | $0.00 \pm 0.00$ a |
| GAL15 | $0.00 \pm 0.00$ a | $0.00 \pm 0.00$ a | $0.00 \pm 0.00$ a | $0.00 \pm 0.00$ a |
| Cyanobacteria | $0.00 \pm 0.00$ a | $0.00 \pm 0.00$ a | $0.00 \pm 0.00$ a | $0.00 \pm 0.00$ a |
| Planctomycetes | $0.01 \pm 0.00$ ab | $0.00 \pm 0.00$ ab | $0.01 \pm 0.00$ a | $0.00 \pm 0.00$ b |
| Patescibacteria | $0.01 \pm 0.00$ a | $0.01 \pm 0.01$ a | $0.01 \pm 0.00$ a | $0.00 \pm 0.00$ a |
| Entotheonellaeota | $0.00 \pm 0.00$ a | $0.00 \pm 0.00$ a | $0.00 \pm 0.00$ a | $0.00 \pm 0.01$ a |
| Armatimonadetes | $0.00 \pm 0.00$ a | $0.00 \pm 0.00$ a | $0.00 \pm 0.00$ a | $0.00 \pm 0.00$ a |
| WPS-2 | $0.00 \pm 0.00$ a | $0.00 \pm 0.00$ a | $0.01 \pm 0.01$ a | $0.00 \pm 0.00$ a |
| Zixibacteria | $0.00 \pm 0.00$ a | $0.00 \pm 0.00$ a | $0.00 \pm 0.00$ a | $0.00 \pm 0.00$ a |
| Dependentiae | $0.00 \pm 0.00$ a | $0.00 \pm 0.00$ a | $0.00 \pm 0.00$ a | $0.00 \pm 0.00$ a |

Note: The data are mean $\pm$ standard error (SE), different letters indicate significant levels ($p < 0.05$).

**Table A2.** Comparison of relative abundance of bacterial class with significant differences among different samples in the Taojia River Basin.

| Taxon | VL | WL1 | WL2 | CL |
|---|---|---|---|---|
| Deltaproteobacteria | $0.11 \pm 0.04$ a | $0.08 \pm 0.03$ ab | $0.04 \pm 0.02$ b | $0.11 \pm 0.03$ a |
| Actinobacteria | $0.07 \pm 0.02$ b | $0.09 \pm 0.01$ b | $0.19 \pm 0.05$ a | $0.08 \pm 0.05$ b |
| Anaerolineae | $0.10 \pm 0.08$ a | $0.07 \pm 0.08$ a | $0.01 \pm 0.01$ a | $0.11 \pm 0.05$ a |
| Alphaproteobacteria | $0.10 \pm 0.03$ b | $0.09 \pm 0.02$ b | $0.14 \pm 0.02$ a | $0.08 \pm 0.02$ b |

**Table A2.** *Cont.*

| Taxon | VL | WL1 | WL2 | CL |
|---|---|---|---|---|
| Thermoleophilia | 0.06 ± 0.06 a | 0.10 ± 0.07 a | 0.15 ± 0.10 a | 0.06 ± 0.08 a |
| Gammaproteobacteria | 0.12 ± 0.02 a | 0.07 ± 0.01 b | 0.03 ± 0.02 c | 0.08 ± 0.03 ab |
| Gemmatimonadetes | 0.04 ± 0.01 a | 0.05 ± 0.02 a | 0.03 ± 0.01 a | 0.04 ± 0.01 a |
| Acidobacteriia | 0.05 ± 0.02 a | 0.07 ± 0.07 a | 0.08 ± 0.04 a | 0.05 ± 0.03 a |
| NC10 | 0.03 ± 0.01 a | 0.03 ± 0.02 a | 0.01 ± 0.01 a | 0.03 ± 0.00 a |
| AD3 | 0.00 ± 0.00 a | 0.01 ± 0.01 a | 0.02 ± 0.03 a | 0.00 ± 0.00 a |
| Ktedonobacteria | 0.01 ± 0.00 a | 0.02 ± 0.02 a | 0.08 ± 0.09 a | 0.00 ± 0.00 a |
| Subgroup_6 | 0.05 ± 0.01 a | 0.04 ± 0.03 a | 0.02 ± 0.01 a | 0.04 ± 0.02 a |
| Acidimicrobiia | 0.02 ± 0.01 a | 0.03 ± 0.01 a | 0.04 ± 0.01 a | 0.03 ± 0.03 a |
| Chloroflexia | 0.01 ± 0.00 a | 0.02 ± 0.02 a | 0.02 ± 0.02 a | 0.03 ± 0.04 a |
| KD4-96 | 0.04 ± 0.01 a | 0.04 ± 0.02 a | 0.01 ± 0.01 a | 0.04 ± 0.02 a |
| Blastocatellia_(Subgroup_4) | 0.02 ± 0.01 a | 0.03 ± 0.03 a | 0.02 ± 0.02 a | 0.02 ± 0.00 a |
| TK10 | 0.01 ± 0.00 b | 0.02 ± 0.01 ab | 0.03 ± 0.02 a | 0.01 ± 0.00 b |
| MB-A2-108 | 0.01 ± 0.01 a | 0.01 ± 0.01 a | 0.01 ± 0.01 a | 0.01 ± 0.02 a |
| Rubrobacteria | 0.00 ± 0.00 a | 0.00 ± 0.00 a | 0.00 ± 0.00 a | 0.00 ± 0.00 a |
| Subgroup_17 | 0.01 ± 0.00 ab | 0.00 ± 0.00 ab | 0.00 ± 0.00 b | 0.01 ± 0.00 a |

Note: The data are mean ± standard error (SE), different letters indicate significant levels ($p < 0.05$).

**Table A3.** Comparison of relative abundance of fungus phylum with significant differences among different samples in the Taojia River Basin.

| Taxon | VL | WL1 | WL2 | CL |
|---|---|---|---|---|
| Ascomycota | 0.14 ± 0.10 a | 0.42 ± 0.29 a | 0.28 ± 0.17 a | 0.20 ± 0.20 a |
| Mucoromycota | 0.29 ± 0.16 a | 0.18 ± 0.09 a | 0.41 ± 0.10 a | 0.23 ± 0.12 a |
| Basidiomycota | 0.13 ± 0.08 a | 0.15 ± 0.13 a | 0.15 ± 0.02 a | 0.22 ± 0.05 a |
| Chytridiomycota | 0.15 ± 0.15 a | 0.08 ± 0.04 a | 0.01 ± 0.00 a | 0.13 ± 0.09 a |
| Zoopagomycota | 0.00 ± 0.01 a | 0.00 ± 0.00 a | 0.00 ± 0.00 a | 0.00 ± 0.00 a |
| Olpidiomycota | 0.00 ± 0.00 a | 0.00 ± 0.00 a | 0.00 ± 0.00 a | 0.00 ± 0.00 a |
| Blastocladiomycota | 0.00 ± 0.00 a | 0.00 ± 0.00 a | 0.00 ± 0.00 a | 0.00 ± 0.00 a |

Note: The data are presented as mean ± standard error (SE), different letters indicate significant levels ($p < 0.05$).

**Table A4.** Comparison of relative abundance of fungus class with significant differences among different samples in the Taojia River Basin.

| Taxon | VL | WL1 | WL2 | CL |
|---|---|---|---|---|
| Sordariomycetes | 0.11 ± 0.10 a | 0.40 ± 0.30 a | 0.13 ± 0.04 a | 0.17 ± 0.18 a |
| Dothideomycetes | 0.00 ± 0.01 a | 0.01 ± 0.01 a | 0.11 ± 0.18 a | 0.01 ± 0.01 a |
| Glomeromycetes | 0.09 ± 0.08 ab | 0.03 ± 0.03 ab | 0.18 ± 0.13 a | 0.03 ± 0.02 b |
| Agaricomycetes | 0.05 ± 0.03 b | 0.07 ± 0.00 b | 0.13 ± 0.02 a | 0.05 ± 0.03 b |
| Mortierellomycetes | 0.19 ± 0.21 a | 0.10 ± 0.06 a | 0.11 ± 0.07 a | 0.16 ± 0.07 a |
| Chytridiomycetes | 0.15 ± 0.15 a | 0.07 ± 0.03 a | 0.01 ± 0.00 a | 0.12 ± 0.08 a |
| Endogonomycetes | 0.02 ± 0.01 a | 0.04 ± 0.06 a | 0.11 ± 0.17 a | 0.04 ± 0.04 a |
| Pucciniomycetes | 0.06 ± 0.06 a | 0.07 ± 0.13 a | 0.00 ± 0.00 a | 0.12 ± 0.12 a |
| Tremellomycetes | 0.00 ± 0.00 a | 0.00 ± 0.00 a | 0.01 ± 0.01 a | 0.03 ± 0.04 a |
| Calcarisporiellales | 0.00 ± 0.00 a | 0.00 ± 0.01 a | 0.00 ± 0.00 a | 0.00 ± 0.00 a |
| Eurotiomycetes | 0.00 ± 0.00 a | 0.00 ± 0.00 a | 0.00 ± 0.00 a | 0.00 ± 0,00 a |
| Saccharomycetes | 0.00 ± 0.00 a | 0.00 ± 0.00 a | 0.02 ± 0.02 a | 0.00 ± 0.00 a |
| Orbimyces | 0.00 ± 0.00 a | 0.00 ± 0.00 a | 0.00 ± 0.00 a | 0.00 ± 0.00 a |
| Microbotryomycetes | 0.00 ± 0.00 a | 0.00 ± 0.00 a | 0.00 ± 0.00 a | 0.00 ± 0.01 a |
| Ramicandelaberales | 0.00 ± 0.01 a | 0.00 ± 0.00 a | 0.00 ± 0.00 a | 0.00 ± 0.00 a |
| Pezizomycetes | 0.00 ± 0.00 a | 0.01 ± 0.01 a | 0.00 ± 0.00 a | 0.00 ± 0.00 a |
| Olpidiomycetes | 0.00 ± 0.00 a | 0.00 ± 0.00 a | 0.00 ± 0.00 a | 0.00 ± 0.00 a |
| Zoopagomycetes | 0.00 ± 0.00 a | 0.00 ± 0.00 a | 0.00 ± 0.00 a | 0.00 ± 0.00 a |
| Blastocladiomycetes | 0.00 ± 0.00 a | 0.00 ± 0.00 a | 0.00 ± 0.00 a | 0.00 ± 0.00 a |
| Monoblepharidomycetes | 0.00 ± 0.00 a | 0.00 ± 0.00 a | 0.00 ± 0.00 a | 0.00 ± 0.00 a |

Note: The data are mean ± standard error (SE), different letters indicate significant levels ($p < 0.05$).

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
