# Peer review of "Soil Microbial Community Composition and Diversity Analysis under Different Land Use Patterns in Taojia River Basin"

_forests, doi:10.3390/f14051004_

Round 1

Reviewer 1 Report

The manuscript "Soil microbial community composition and diversity analysis under different land use patterns in Taojia River Basin" by Zhe He, et al. aims to evaluate the influence of land use on the composition and diversity of soil bacterial and fungal communities in the Taojia River Basin. 

Indicate and justify the difference between land use pattern and land use. I suggest removing the word "pattern" along the document, including the title. 

Introduction

This section needs to reflect a story instead of a list of previous works related to the topic. Almost all the information can be summarized by a table. 

Materials and Methods

A detailed description of each land use (vegetable land, wasteland, woodland, and cultivated land) must be provided. Specify the date and season of sampling. Also, discuss the relevance of sampling in a determined season.

The use of partial least squares path modelling needs to be better justified, mainly because the debate regarding if its use is appropriate has yet to be resolved. See:

https://doi.org/10.1016/j.jom.2016.05.002

https://doi.org/10.1016/j.leaqua.2010.10.010

https://doi.org/10.1007/s11135-018-0689-6

More than twelve samples must be used to establish a causal relationship between the latent variables of different land use patterns and the diversity of soil microbial communities. I strongly suggest removing the analyses. The overall goodness of fit is insufficient to determine the adequate use of PLS-PM; the authors need to present the heterotrait-monotrait ratio of correlations (HTMT), assess the discriminant validity, and the standardized root mean square residual (SRMR) as an approximate measure of fit. Also, present a bootstrap-based test of overall model fit and indicate the order of the factors in the model. In each case, the fit's acceptance ranges need to be specified. Besides, determine how the authors introduce the land use variable.

Results

So many results are presented without any justification.

Authors must decide what story they are going to tell. Results of bacteria and fungi at the family, class and genus levels are presented. Because of the objective of presenting so many results, they must make a real effort to synthesize and prioritize the information. Most results can be added as supplementary material, leaving only those that serve the diagnosis they propose (phylum?).

Discussion

The authors need to synthesize and define this section according to their objectives. The main objective was to evaluate the influence of land use on the composition and diversity of soil bacterial and fungal communities; instead, a lengthy discussion is given for soil properties (576-605), soil enzymatic activity (not measured in this study; 606-636). Instead, the authors need to highlight and frame how their results will help to reach the goal of providing a scientific theoretical basis for the comprehensive ecological restoration of the Taojia River Basin according to local conditions.

Conclusion

The authors need to highlight the relevance and novelty of their study; what can we learn from this study? Which is the take-home message?

L731-738 – This is no new insight; many studies have reached the same conclusion. What is new or different, or relevant in this case?

Specific comments

L22 – Abbreviations need to be defined in first use. 

L103 and 106 – replace "serious" with a quantitative assessment

L123 Use the World Reference Base for Soil Resources to provide soil classification.

L136 textural composition?

L142 – deposit or specify the Primers sequence used in all cases.

L176 – Specify in the manuscript what the author means by "microbial indicators."

L576-645 - Unnecessary

Figure 1- The Map and figure captions need to be substantially improved. Scale, coordinates, coordinate system, grid reference, etc. Also, mark the 12 plots used. 

Figures need to be self-supported; thus, a better description is needed. Also, include credits to the map's layers.

Table 1. The table exceeds the page limits. It is not possible to see the P values. 

Figures 2 and 4 – A better description is needed in the caption (what are we looking at in this figure?). 

Figures 3 and 5 – A better description is needed in the caption. Define abbreviations. 

Fig. 6 and 8 (and figures related)- The Y-axis's label is enormous compared to the figure size. Define abbreviations. 

In all figures- use coherent abbreviations for the land uses.

Table type 2. Use 0.00 instead of 0.

Is there a particular interest in reporting a missing taxon in all land uses?

The authors show letters to indicate significant differences (or similarities) even in these taxa with 0 abundance. Check the data coherence of, for example, Planctomycetes.

Author Response

Responses to Reviewers’ Comments

We are grateful to thank you all, the editor and anonymous reviewers, for the valuable suggestions and comments on our manuscript entitled “Soil microbial community composition and diversity analysis under different land use patterns in Taojia River Basin” (Manuscript ID forests-2305737).

We have taken seriously all the questions, suggestions, and comments raised by the reviewers. The following are the point-by-point replies, with reference to the order of the reviewers’ comments (black italic type), and the changes have also been revised in the submitted manuscript. The modified section has been marked in yellow.

Comments and Suggestions for Authors 2

The manuscript "Soil microbial community composition and diversity analysis under different land

use patterns in Taojia River Basin" by Zhe He, et al. aims to evaluate the influence of land use on

the composition and diversity of soil bacterial and fungal communities in the Taojia River Basin.

Response: Thanks for the positive comments on our manuscript. Based on your comments, we have revised the introduction, materials, results, discussion, and analysis, and we have carefully revised our manuscript point-by-point as follows:

(1) Indicate and justify the difference between land use pattern and land use. I suggest removing the

word "pattern" along the document, including the title.

.

Response: Thanks for the your comments on land use types and land use, Land use pattern refers to the spatial arrangement of land use types within a particular area or region, whereas land use refers to the human activities or functions that occur on a particular piece of land. The difference between the two is that land use pattern is concerned with the overall spatial organization of land use  in a particular area,  while land use is concerned with the specific activities and functions that take place on individual parcels of land, While land use is concerned with the specific activities and functions that take place on individual parcels of land, Thanks to the editors for their suggestions, we have considered the content of this article and decided to keep using patterns.

(2) This section needs to reflect a story instead of a list of previous works related to the topic. Almost

all the information can be summarized by a table.

Response: Thank you for the content you proposed in the introduction. After reading other literature, we have partially modified the narrative logic and content of the introduction. Please see Lines 44, 57, 69-74, 76-80, 86-90, 97-112.

(3)A detailed description of each land use (vegetable land, wasteland, woodland, and cultivated

land) must be provided. Specify the date and season of sampling. Also, discuss the relevance of

sampling in a determined season..

Response: Thank you very much for your suggestions, We described the uses of the four plots in detail, mentioned the sampling season and date in the paper, and discussed the influence of the season on soil microorganisms in the discussion section. Please see Lines 120-124,552-562.

(4)The use of partial least squares path modelling needs to be better justified, mainly because the debate regarding if its use is appropriate has yet to be resolved. See: https://doi.org/10.1016/j.jom.2016.05.002 https://doi.org/10.1016/j.leaqua.2010.10.010 https://doi.org/10.1007/s11135-018-0689-6. More than twelve samples must be used to establish a causal relationship between the latent variables of different land use patterns and the diversity of soil microbial communities. I strongly suggest removing the analyses. The overall goodness of fit is insufficient to determine the adequate use of PLS-PM; the authors need to present the heterotrait-monotrait ratio of correlations (HTMT), assess the discriminant validity, and the standardized root mean square residual (SRMR) as an approximate measure of fit. Also, present a bootstrap-based test of overall model fit and indicate the order of the factors in the model. In each case, the fit's acceptance ranges need to be specified. Besides, determine how the authors introduce the land use variable.

Response: Thank you very much for your suggestions on PLSPM structure model. After carefully reading the literature provided by the editor, we searched for relevant materials and listed the advantages of PLS-PM: PLS-PM is more suitable for small sample data and non-normal data. Because PLS-PM uses partial least square method, it has low requirements on data distribution and sample size, while SEM parameter estimation has high requirements on sample size and data distribution. PLS-PM can use repeated adaptive selection in model fitting to improve the interpretability of the model and eliminate the interference of collinearity, noise and other factors. However, SEM requires more assumptions such as data migration and collinearity, and its parameter estimation may be affected by the multicollinearity problem. When there are multiple potential variable groups in the model, PLS-PM is easier to extract group features and establish association structure. SEM, on the other hand, may require the determination of potential variable groups before modeling. In addition, more than 12 samples were used to construct the PLS-PM equation in this study, and the overall goodness of fit was greater than 0.7, indicating that the overall explanatory ability of the structural model was good. All the equation models in this study were feasible and credible, while HTMT and SRMR were not applicable in this equation model. After the initial construction of the structural equation model, Observe the loading value, delete the factors less than 0.7, and construct new ones. The value of R2 indicates the ability of the intrinsic latent variable to explain other intrinsic latent variables. If the loading value is greater than or equal to 0.6, it is better than 0. 3 moderate, less than 0.3 poor. In this study, the introduction of land use types was carried out by digitizing dependent variables and importing them into the equation model for further analysis.

(5)Authors must decide what story they are going to tell. Results of bacteria and fungi at the family,

class and genus levels are presented. Because of the objective of presenting so many results, they must make a real effort to synthesize and prioritize the information. Most results can be added as supplementary material, leaving only those that serve the diagnosis they propose (phylum?).

Response:Thanks for your important comments on the manuscript, we have deleted the genus classification in the results, effectively combined the Genus and phylum, presented the results better and more clearly, and placed most of the rest in supplementary materials. Please see Lines 297-325,373-396.

(6)The authors need to synthesize and define this section according to their objectives. The main

objective was to evaluate the influence of land use on the composition and diversity of soil bacterial and fungal communities; instead, a lengthy discussion is given for soil properties (576-605), soil enzymatic activity (not measured in this study; 606-636). Instead, the authors need to highlight and frame how their results will help to reach the goal of providing a scientific theoretical basis for the comprehensive ecological restoration of the Taojia River Basin according to local conditions.

Response: Thank you for your important comments on my manuscript. According to our research objectives, we comprehensively analyzed the impact of land use patterns on soil microorganisms, and according to the results, discussed and studied the important factors affecting soil microorganisms and the impact of the results on the ecological environment. Please see Lines 477-497, 501-509, 524-532, and 536-564 of discussion in the revised manuscript.

(7)The authors need to highlight the relevance and novelty of their study; what can we learn from this study? Which is the take-home message? L731-738 - This is no new insight; many studies have reached the same conclusion. What is new or different, or relevant in this case?

Response: Thank you for your comments on my manuscript. According to the above analysis results, we extracted and elaborated the key information, and had a deeper and more specific discussion on the repetitive conclusions raised by the editor.Please see Lines 569-574, 587-592of conclusions in the revised manuscript.

(8)L22 - Abbreviations need to be defined in first use.

Response: Thank you for the comments on our manuscript. We have adjusted the problem of Abbreviations need to be defined in first use. Please see Lines 22 of abstract in the revised manuscript.

(9)L103 and 106 - replace "serious" with a quantitative assessment

Response: Thanks for you advice. We use concrete, quantitative data instead of the word "serious,". Please see Lines 97-99 of introduction in the revised manuscript.

(10)L123 Use the World Reference Base for Soil Resources to provide soil classification.

Response: Thanks for this valuable suggestion. Based on the your advice, we reclassified three types of land by looking up USDA soil Taxonomy. Please see Lines 119-120 in the revised manuscript.

(11)L136 textural composition?

Response: Thanks for your suggestion, we replaced mechanical composition with textural composition in this paper. This is our article more accurate.

(12) L142 – deposit or specify the Primers sequence used in all cases.

Response: Thanks for these valuable suggestions. We Insert reference for primers set and PCR condition. Please see Lines 148-150, 156-158, and 160-161 in the revised manuscript.

(13) L176 - Specify in the manuscript what the author means by "microbial indicators.".

Response: Thanks for your suggestion, In this paper, we have refined the microbiological indicators to ensure that readers can understand more intuitively.

(14)L576-645 - Unnecessary

Response: Thanks for this valuable suggestion. This part of the article has been deleted

(15)Figure 1- The Map and figure captions need to be substantially improved. Scale, coordinates,

coordinate system, grid reference, etc. Also, mark the 12 plots used. Figures need to be self-supported; thus, a better description is needed. Also, include credits to

Response: Thanks for this valuable suggestion. According to the request, the map was remade and 12 sample sites were marked. Please see Fig1 in the revised manuscript.

(16)Table 1. The table exceeds the page limits. It is not possible to see the P values..

Response: Thanks to the editor's advice, we have revised the picture and corrected the title. Please see Table1 in the revised manuscript.

(17)The Figures 2 and 4 - A better description is needed in the caption (what are we looking at in this figure?). Figures 3 and 5 - A better description is needed in the caption. Define abbreviations. Fig. 6 and 8 (and figures related)- The Y-axis's label is enormous compared to the figure size. Define abbreviations. In all figures- use coherent abbreviations for the land uses.

Response: Thanks for the editor's suggestions on picture title modification, supplementary materials and legend standardization. We have made modifications one by one according to the editor's suggestions. Please see Fig 2, Fig 3, Fig 4, Fig 5, Fig 6, Fig 7, Fig 8, Fig 9, Fig 10, Fig A1, Fig A2, Fig A3, Fig A4, Table A1, Table A2, Table A3, and Table A4 in the revised manuscript.

(21)Is there a particular interest in reporting a missing taxon in all land uses?

Response: Thanks to the editor's comments, we have added data on the missing categories in the supplementary materia.Please refer to the supplementary materials.

(22)The authors show letters to indicate significant differences (or similarities) even in these taxa with 0 abundance. Check the data coherence of, for example, Planctomycetes.

Response: Thanks for the editor's opinion. Since the relative content of taxonomic group with zero abundance shown in the table is lower than 0.005, the form of presentation is 0 in the process of data statistical analysis, which is not significant.

Reviewer 2 Report

Dear Editor and Authors,

Manuscript ID forests-2305737 has scientific merit and addresses a topic of interest to Journal Forest (MDPI) readers. However, the manuscript has some shortcomings that authors should consider before it is considered for publication.

There are several comments in the pdf file that will guide authors in improving the manuscript.

The article has a very large volume of figures (22) and tables (7), the authors need to synthesize the results and be objective in the presentation of information. Fungal community data needs to be revised. This is because two phyla of green algae are in the analysis as belonging to the fungi group.

My recommendation is a major revision.

Author Response

Responses to Reviewers’ Comments

We are grateful to thank you all, the editor and anonymous reviewers, for the valuable suggestions and comments on our manuscript entitled “Soil microbial community composition and diversity analysis under different land use patterns in Taojia River Basin” (Manuscript ID forests-2305737).

We have taken seriously all the questions, suggestions, and comments raised by the reviewers. The following are the point-by-point replies, with reference to the order of the reviewers’ comments (black italic type), and the changes have also been revised in the submitted manuscript. The modified section has been marked in yellow.

Comments and Suggestions for Authors 1

Manuscript ID forests-2305737 has scientific merit and addresses a topic of interest to Journal Forest (MDPI) readers. However, the manuscript has some shortcomings that authors should consider before it is considered for publication. There are several comments in the pdf file that will guide authors in improving the manuscript. The article has a very large volume of figures (22) and tables (7), the authors need to synthesize the results and be objective in the presentation of information. Fungal community data needs to be revised. This is because two phyla of green algae are in the analysis as belonging to the fungi group.

Response: Thanks for the positive comments on our manuscript.We reprocessed, analyzed, and discussed the fungal data, and analyzed the community composition of bacteria and fungi at the taxonomic level of phyla and class according to the editorial opinion. Based on your comments, we carefully revised our manuscript point-by-point as follows:

(1) The Streptophyta includes six main lineages of freshwater green algae. https://www.sciencedirect.com/topics/agricultural-and-biological-sciences/streptophyta It's not fungus. Although it belongs to the Eukaryote domain.

Response: Thanks for your pertinent suggestions and comments, which are very significant in improving our manuscript. We re-processed and analyzed data from the fungi community, Please see Lines 265-278, 279-291of soil fungal community diversity analysis in the revised manuscript.

Lines 265-278: After optimization, filtration and removal of low-quality sequences, a total of 178,049 effective sequences were obtained from 12 sequenced soil samples, with an average of 14,837 valid sequences per soil sample. According to the 97% sequence similarity, the OTUs detected in vegetable land, wasteland, woodland, and cultivated land soil were 5,577, 8,762, 8,414, and 3,093, respectively. Among them, the total number of OTUs is 9,615, and the unique OTUs are 989, 2,566, 1,783, and 3, respectively. The number of common OTUs in soil under the four land use patterns was 2,767. The three land use patterns of vegetable land, woodland, and cultivated land have no number of OTUs in common, and the common OTUs number of wasteland, woodland and cultivated land is also 0. Meanwhile, there is no common OTU number between cultivated land and vegetable land or between cultivated land and woodland. Woodland soils have the fewest endemic OTUs, only 3, which is much smaller than the endemic OTUs in wasteland land soils, 2,566 (Fig. A2).

Lines 279-291: while the soil pH, soil water content, soil organic carbon, total nitrogen, total phosphorus, sucrase, urease, acid phosphatase, catalase, and microbial biomass carbon, nitrogen, and phosphorus showed higher in 7-yr-old stand than other two stands of CFPIn the soils of the study area, the Shannon index, observed species index , and Chao1 index of fungal species in woodland soil were the highest, which were 3.95, 186.67, and 184.25, respectively, and the lowest in cultivated soil, at 3.08, 137.33, and 139.58, respectively, followed by wasteland and vegetable land. The Simpson’s index of wasteland soil was the lowest, 0.81; the Simpson index of woodland soil was smaller than that of vegetable soil; the Simpson index of cultivated soil was the highest, 0.96, showed that the dominant species had the greatest impact on the overall species diversity in cultivated land, with species diversity being relatively low. However, a few dominant species in the wasteland had the least effect on the overall species diversity, which was relatively high.The results indicating that the species richness of fungi in woodland soil was the highest, the species richness of fungi in cultivated soil were the lowest, followed by wasteland and vegetable land. (Fig. 3)         .

(2) Chlorophyta:Green algae constitute the most heterogeneous group of photoautotrophic protoctists inhabiting the biosphere and show an enormously wide variability of shape, size, and habit. As primary producers, green algae have an importance on our planet comparable to that of rainforests. https://www.sciencedirect.com/topics/agricultural-and-biological-sciences/chlorophyta It's not fungus. Although it belongs to the Eukaryote domain.

Response: Thanks for your suggestion. We We analyzed soil fungal community composition and its correlation with soil physicochemical properties. Please see Lines 373-385, 386-398 of soil fungal community composition analysis in the revised manuscript.

Lines 373-385: Through sequencing data analysis, the microorganisms in a total of 12 soil samples in the four different land use patterns of vegetable land, wasteland, woodland, and cultivated land can be divided into 7 phyla, 40 classes, 138 families, 155 genera, and 113 species. At the phylum level, the dominant phyla of vegetable land, woodland, and cultivated land were Mucoromycota, 29.39%, 41.36%, and 22.67%, respectively. Ascomycota is the most abundant in wasteland, at 42.16%. Phyla with an average relative abundance of more than 10% include  Basidiomycota (13.18%~22.28%) and Chytridiomycota (1.01%~14.76%) (Fig. 7). The level of fungal phyla did not differ significantly across land use patterns (p>0.05). The differences in fungal abundance of specific phyla are shown in Table A3. The dominant phylum Mucoromycota had the greatest difference in abundance between woodland and wasteland, Ascomycota had the greatest difference in abundance between wasteland and vegetable land.

Lines 386-398: At the class level, the main fungal taxa with an  relative abundance greater than 10% that were detected in vegetable land, wasteland, woodland, and cultivated land were Sordariomycetes (11.02%~39.99%) Mortierellomycetes (10.27%~18.73%). Sordariomycetes had the largest relative abundance in wasteland and cultivated land, 18.73% and 16.92%, respectively. Mortierellomycetes is the most abundant in vegetable land, at 24.88%; Glomeromycetes had the largest relative abundance in woodland, which was 18.38% (Fig. 7). Blastocladiomycetes (0.18%) were found only in vegetable land soil. Among the woodlands, Glomeromycetes and Agaricomycetes showed the highest relative abundance, which was significantly different from the other three land use patterns (p<0.05). The differences in fungal abundance in specific phyla are shown in Table A4.

(3) Display numerical values of the results. This draws the reader's attention to the manuscript. Insert a sentence highlighting the conclusion of the study or prospects for future research in this field of study.

Response: Thanks for your suggestion. We performed RDA analysis of soil fungal community composition reproduction. Please see Lines 415-433 of soil fungal community composition analysis in the revised manuscript.

Lines 415-433: RDA was used to explore the relationship between soil fungal community composition and soil physicochemical properties under four land use patterns. This analysis showed (Fig. 9) that RDA1 and RDA2 explained 31.04% and 25.61% of soil fungal community composition, respectively. The fungal community composition of woodland soil was more similar than that of other land use patterns. TN was positively correlated with moisture content, there was a negative correlation between pH and moisture content. Soil clay and moisture content had a greater effect on fungal community composition, Silt had the least effect on fungal community composition. Among them, pH was positively correlated with fungal community composition  in wasteland and negatively correlated with fungal community composition in vegetable land, cultivated land, and woodland. Soil TN and moisture content were positively correlated with soil fungal community composition in vegetable land, cultivated land, and woodland, but negatively correlated with soil fungal community composition in wasteland. The dominant fungal phylum was associated with soil properties such as clay, pH, silt, moisture content, and TN. For example, the abundance of Ascomycota in wasteland increased with increasing soil pH and increased with decreasing soil Silt and Clay. With the decreases of soil TN and moisture content, the relative abundance of Mucoromycota and Basidiomycota decreased in the four land use patterns.

(4)Use the soil classification system (Soil taxonomy Staff - USDA or World reference base - FAO): https://www.nrcs.usda.gov/resources/guides-and-instructions/soil-taxonomy https://www.fao.org/soils-portal/data-hub/soil-classification/world-reference-base/en/

Response: Thanks for this valuable suggestion. Based on the your advice, we reclassified three types of land by looking up USDA soil Taxonomy. Please see Lines 119-120 in the revised manuscript.

Lines  119-120: The soil is mainly Ultisol, Oxisol, and Alfisol [22]

(5)Insert reference for primers set and PCR condition.

Response: Thanks for these valuable suggestions. We Insert reference for primers set and PCR condition. Please see Lines 148-150, 156-158, and 160-161 in the revised manuscript.

Lines 148-150: The bacterial primer sequence were 515F and 806R [27,28]. (515F: 5'-GTGCCAGCMGCCGCGG-3′; 806R: 5′-GGACTACHVGGGTWTCTAAT-3′). PCR amplification system (50 μL) [29]: 2 × Taq master Mix 25 μL, Bar-PCR upstream and downstream primers (10 μmol/L) 1 μL, template DNA 1 μL, ddH2O to 50 μL.

Lines 156-158: NS1 and GCFung [30,31]. NS1 (5′-GTAGTCATATGCTTGTCTC-3′) GCFung (5′-GC clamp-ATTCCCCGTTACCCGTTG-3′). PCR amplification system (50 μL) [32]

Lines 160-161:Each sample was amplified in triplicate with the 50- μl reaction mixtures under the following conditions [33]

(6)How and where were the DNA sequences processed?Present a brief description of sequence processing (main procedures, packages or programs used, cut and alignment criteria).

Response: Thank you very much for this advice, which has helped the science and reliability of the manuscript, and we have increased the processing process of DNA sequence. Please see Lines 169-183, 184-192 in the revised manuscript.

Lines 169-183: The Illumina TruSeq Nano DNA LT Library Prep Kit was used to prepare the sequencing library. The highlight base at the 5 'End of the amplified DNA sequence was excised, a phosphate group was added, and the missing base at the 3' end was completed by End Repair Mix 2 in the kit. A sequencing adapter containing a library specific tag was added at the 5 'end of the sequence. BECKMAN AMPure XP Beads were used to screen and purify the library system. The DNA fragments attached to the adaptor were amplified by PCR, and the library enrichment products were purified again using BECKMAN AMPure XP Beads. The final fragment selection and purification of the library was performed by 2% agarose gel electrophoresis. The Agilent High Sensitivity DNA Kit and Quant-iT PicoGreen dsDNA Assay Kit were used to quality check the library and quantitate the library on Promega QuantiFluor system. The qualified sequencing libraries were diluted in gradient, mixed in the corresponding proportion according to the required sequencing amount, denatured with NaOH, and sequenced.

Lines 184-192: The paired-end raw reads were proceeded using Quantitative Insights Into Microbial Ecology (QIIME) pipeline [34] for quality filtering, trimming, and chimera checking. After quality checking and noise reduction, FLASH (Version 1.2.11) and Usearch (Version 10) were used to cluster the data. Operational Taxonomic units (OTUs) were clustered according to the criterion of 97% similarity, and the resulting otus represented sequences [35, 36]. Finally, Mothur method and the SSUrRNA database [37]of SILVA1323 [38] were used for species annotation analysis of OTUs sequencesand the microbial community structure was counted at each taxonomic level.

(7)Remove Streptophyta and Chlorophyta sequences (Green Algae). Rewrite the results from the new figures without the green algae sequences.

Response: Thanks for the your comments, which helped us find the problems in the article in time and correct them. Meanwhile, we rewrote the content of the fungus part. Please see Lines 446-465, 494-499, 501-509, 516-522, 536-564, 569-574, and 574-592 of results, discussion, and conclusions in the revised manuscript.

Lines 446-465: The goodness-of-fit (GOF) of the model was 0.737, indicating that the overall explanatory ability of the model was a good measure.It can be seen that land use pattern has a positive effect on soil textural composition and soil humidity, and has a very significant impact on soil humidity (p<0.01) and a negative impact on soil chemical properties. Soil textural composition has a  significant positive effect on soil chemical properties (p<0.01) and a very significant negative effect on soil humidity (p<0.001), while soil humidity has a extremely significant positive effect on soil chemical properties(p<0.001). Soil textural composition, soil moisture, and soil chemical properties all had effects on bacterial and fungal community diversity, but none of them was significant. The explanatory degree (R2) of soil bacteria and soil fungi was 0.26 and 0.73, respectively, indicating that fungi were more affected by different soil physicochemical factors than bacteria, that is, different land use patterns had greater effects on soil fungal community diversity than on soil bacterial community diversity. Effect represents the influence state of one variable on another variable, soil humidity and soil chemical properties  have a positive total effect on soil bacteria and fungal community diversity, soil textural composition has negative effects on fungal and bacterial community diversity. Land use pattern has a positive total effect on soil bacterial community diversity, and has a negative total effect on soil fungal community diversity, soil bacterial community diversity has effect on soil fungal community diversity (Fig. 10).

Lines 494-499: In vegetable land, woodland, and cultivated land, the main dominant fungi phyla was Mucoromycota, the relative abundance of Ascomycota was highest in wasteland. The dominant phylum Mucoromycota had the greatest difference in abundance between woodland and wasteland, Ascomycota had the greatest difference in abundance between wasteland and vegetable land,

Lines 501-509: This may be because different fungi differ in their ability to efficiently decompose organic matter in soil. The results of Spearman analysis showed that the dominant groups in the bacterial and fungal communities had a significant correlations with soil pH, clay, and sand, These results indicated that soil texture composition has an important effect on soil microbial community composition in river basin. The RDA results showed that soil clay, pH, and moisture were the key environmental factors affecting soil microbial communities.It is inconsistent with previous research results that soil TN, TC and other components play a key role in soil microbial community composition[43,44].

Lines 516-522: Different land use patterns not only affect the difference of microbial community structure, but also affect the diversity of soil microbial community [39]. In this study, the effect of land use patterns on the diversity of soil microbial communities showed significant differences under the four different land use patterns. The results showed that the bacterial diversity of vegetable land was higher than that of other land uses, and the fungal diversity of woodland was also significantly higher than that of other land uses.

Lines 536-564: while they have a significant negative impact on soil moisture content, Meanwhile, soil moisture content has a significant positive impact on soil chemical properties. Soil chemical properties include pH, TN, TC, etc. Soil pH directly or indirectly affects the diversity of soil microorganisms [49]. In this study, it was found that soil TC and TN have a smaller impact on soil diversity, which is different from previous studies that found soil nutrients have a greater impact on the diversity and abundance of soil microbial communities [50,51]. The lowest diversity of bacteria in barren land may be due to the lack of nutrients and organic matter in the soil, lack of coverage, long-term exposure of the soil surface to sunlight, and extreme dryness of the soil, which hinder the survival and reproduction of bacteria. while the lowest fungal diversity in cultivated land may be  influenced by humidity, pH, and TN, while seasonal factors also need to be considered, which is similar to the conclusion of Ji Chuning et al. that pH value and water content have significant effects on soil fungal community diversity in the study of fungal community succession of reclaimed soil in mining areas [52]. Many factors, such as seasonal variation, land use and soil management have a significant impact on bacterial richness and diversity [53,54]. The sampling time is in the spring, and the spring is the natural growth of plants, different vegetation differences, will make the physicochemical properties of soil changes, thus affecting soil microorganisms. Human's agricultural management measures to different areas in spring and the change of climate environment in spring may change the soil characteristics of the sample land, and then affect soil microorganisms.Soil texture composition, soil moisture, and soil chemical properties all have an impact on the diversity of soil fungi and bacterial communities. Aside from soil texture composition negatively impacting soil microbial community diversity, other soil physical and chemical properties overall have a positive impact on soil microbial community diversity. Therefore, it can be seen from the above results that soil physicochemical properties are the main factors that affect microbial diversity and community structure. [55-56].

Lines 569-574: The results of Spearman analysis showed that the dominant groups in the bacterial and fungal communities had a significant correlations with soil pH, clay, and sand, These results indicated that soil texture composition has an important effect on soil microbial community composition in river basin. The RDA results showed that soil clay, pH, and moisture were the key environmental factors affecting soil microbial communities.

Lines 574-592: The relative abundance of Proteobacteria, Actinobacteria was the highest in the bacterial communities and Mucoromycota and Ascomycota was the highest in the fungi. Through the analysis of alpha diversity, it was known that bacteria and fungi had the highest species richness in vegetable land and woodland, respectively. The construction of PLS-PM demonstrated that soil fungi were more affected by land use patterns than soil bacteria in the Taojia River Basin. Based on the above results, it is speculated that woodland fungal communities in river regions may play an important role in the whole ecosystem. Therefore, forest management and protection are the key to soil ecological restoration in the Taojia River Basin. In summary, different land use patterns in the Taojia River Basin determine all aspects of soil microorganisms, including soil physicochemical properties, microbial diversity and community composition, and the correlation of soil environmental factors. In most of the previous studies, the effects of soil texture composition on soil microbial community composition and diversity were not significant. Therefore, this study has important theoretical and practical significance for revealing more internal relationships between land use types and soil microbial diversity and community composition, analyzing soil microbial environment, and maintaining ecosystem stability.

(8)Display a maximum of 2 decimal places.

Response: Thank you for this advice We reworked the decimal point problem. Please see Lines 280-285 in the revised manuscript.

Lines 280-285: which were 3.95, 186.67, and 184.25, respectively, and the lowest in cultivated soil, at 3.08, 137.33, and 139.58, respectively, followed by wasteland and vegetable land. The Simpson’s index of wasteland soil was the lowest, 0.81; the Simpson index of woodland soil was smaller than that of vegetable soil; the Simpson index of cultivated soil was the highest, 0.96

(9)Does the heatmap scale represent the Z-score?The figure should be self-explanatory. Figure titles are too short and without a good description.There is overlapping information: figure 6, table 2 and figure 7. Standardize the legend as suggested: VL = vegetable land WL1 = wasteland WL2 = woodland CL = cultivated land

Response: Thanks for this valuable suggestion. We find the Z-score is a measure of how many standard deviations a value is from the mean, and it is often used in data analysis to standardize values so that they can be compared across different samples. In a heatmap, the Z-score scale can be used to show the relative expression levels of different genes or proteins, According to the your opinion that the content is repeated, we deleted the picture and combined its related analysis with the above content. The specific modification has been repeated in the above part, please refer to the modified version for details.

(10)Explore the bacterial community at only 2 taxonomic levels. For example, phylum and class; Phylum and Family; Family and genus.If you choose to explore three levels, one of the levels must be added to the Supplementary Material.

Response: Thanks to the your valuable comments, we re-analyzed the data, deleted the species composition at genus level, and integrated the relative abundance map at phylum and class level, which greatly reduced the text of the article and helped the visualization of information, The specific modification has been repeated in the above part, please refer to the modified version for details.

(11)Green algae! Rephrase the topic as you remove the algae sequences and reanalyze the data. See considerations for the bacteria community. Make the changes and submit. The manuscript is long. There is a lot of overlap in information and analysis. There are 22 figures and 7 tables in the manuscript. The authors need to be objective in presenting the results. It is tiring to review an article with a very large volume of overlapping information.

Response: Thank you for the invaluable comment on our manuscript. We have timely corrected the duplication, jumble and fungal data errors of the article put forward by the editor, and the manuscript has been timely corrected. There are duplicates in the modified content, and all of them are presented in the revised version. Thanks again for all the comments put forward by the editor.

(12)article?

Response: Thank you for you comment. We modified acticle to study. Please see Lines 17 in the revised manuscript.

z

(13)avoid abrevitation in abstract.

Response: Thank you for you comment. We have corrected the abbreviation that was first used in the definition. Please see Lines 22 in the revised manuscript.

Lines 22: and a partial least squares path model (PLS-PM)

(14)the repeated placement of the two words seems odd.replace "different land use" by "land use" only.

Response: Thank you for the invaluable comment on our manuscript. Please see Lines 25-26 in the revised manuscript.

Lines 25-26: Proteobacteria is the dominant phylum (20.69%-32.70%)

(15)Display numerical values of the results. This draws the reader's attention to the manuscript. Insert a sentence highlighting the conclusion of the study or prospects for future research in this field of study.

Response: Thanks to the your valuable comments on our manuscript. We present the data of the research results in the abstract part, and add the conclusion and research significance of this study at the end Please see Lines 25-38 in the revised manuscript.

Lines 28-38: Proteobacteria is the dominant phylum (20.69%-32.70%), and Actinobacteria is the dominant class (7.99%-16.95%). The species richness of fungi in woodland was the highest, while was the lowest in cultivated land. The dominant phylum of fungi in vegetable land, woodland, and cultivated land is Mucoromycota, 29.39%, 41.36%, and 22.67%, respectively. Ascomycota (42.16%) is the dominant phylum in wasteland. Sordariomyetes of Ascomycota is the dominant class in wasteland and cultivated land. Mortierellomycetes and Glomeromycetes of Mucoromycota are the dominant class in vegetable land and woodland. The results of Spearman analysis showed that the dominant groups in the bacterial and fungal communities had a significant correlations with soil pH, clay, and sand (p<0.01). The RDA results showed that soil clay, pH, and moisture were the key environmental factors affecting the diversity of soil microbial communities. Fungal diversity is more affected by different land use patterns than bacteria. These results provided a theoretical basis for the changes in soil microbial community composition and diversity in river basins.

(16)Present more clearly the objectives of the study.

Response: Thanks for your valuable comments on our manuscript, we have added this research objective at the end of the introduction. Please see Lines 99-112 in the revised manuscript.

Lines 99-112: In view of this, this study used high-throughput sequencing methods to analyze and compare the bacterial and fungal community composition and diversity as well as their related environmental factors under different land use patterns in the Taojia River Basin. The main purposes of this study were: (1) to investigate the response of soil microbial diversity to different land use patterns; (2) to reveal the microbial community composition and soil physicochemical properties under different land use patterns; and (3) to evaluate the influence of soil physicochemical properties on microbial community composition and diversity. This study will provide a theoretical basis for the changes in soil microbial community composition and diversity in river basins, and has important practical implications for the comprehensive analysis of soil microbial diversity and the implementation of land-use management tailored to the characteristics of the Taojia River Basin, including soil ecological restoration and maintenance of sustainable land use.

(17)Abbreviation suggestion: VL = vegetable land;WL1 = wasteland;WL2 = woodland;CL = cultivated land

Response: Thanks especially to the editor for his comments on my manuscript, we abbreviated all the samples in the manuscript. Please see revised manuscript.

(18)Insert references to the methodologies used for pH, macro and micronutrient contents.

Response: Thanks to the editor's comments, we have supplemented the corresponding references. Please see revised manuscript.

  • Table 1 is offset. Part of the table is outside the margin of the page (unreadable).The table title should be at the top.

Response: Thanks to the editor's advice, we have revised the picture and corrected the title. Please see Table1 in the revised manuscript

  • Remove from manuscript or add to supplementary material.Suggestion:Figure should change name; Figure should be transferred to the supplementary material.

Response: Thanks for the editor's suggestions on picture title modification, supplementary materials and legend standardization. We have made modifications one by one according to the editor's suggestions. Please see Fig 2, Fig 3, Fig 4, Fig 5, Fig 6, Fig 7, Fig 8, Fig 9, Fig 10, Fig A1, Fig A2, Fig A3, Fig A4, Table A1, Table A2, Table A3, and Table A4 in the revised manuscript.

Round 2

Reviewer 1 Report

The authors followed all the suggestions made. I have no further comments. Only Editorial work is needed to homogenize the presentation of the results in both text (see blue text) and figure and tables captions (no complete stand). 

Author Response

Responses to Reviewers’ Comments

We are grateful to thank you all, the editor and anonymous reviewers, for the valuable suggestions and comments on our manuscript entitled “Soil microbial community composition and diversity analysis under different land use patterns in Taojia River Basin” (Manuscript ID forests-2305737).

We have taken seriously all the questions, suggestions, and comments raised by the reviewers. The following is the main content of this modification, with reference to the order of the reviewers’ comments (black italic type), and the changes have also been revised in the submitted manuscript. The modified section has been marked in yellow.

Comments and Suggestions for Authors 1

The authors followed all the suggestions made. I have no further comments. Only Editorial work is needed to homogenize the presentation of the results in both text (see blue text) and figure and tables captions (no complete stand). 

Response:Thanks for the editor's suggestions. According to the suggestions, we have unified processing and improvement of text, figure and table captions.

Reviewer 2 Report

The manuscript should be considered for publication in this form.

Author Response

Please refer to the attachment for our specific modification reply.